# A developmental increase of inhibition promotes the emergence of hippocampal ripples

Irina Pochinok [1], Tristan M. Stöber [2], Jochen Triesch [2], Mattia Chini [1,3] & Ileana L. Hanganu-Opatz [1,3]

Sharp wave-ripples (SPW-Rs) are a hippocampal network phenomenon critical for memory consolidation and planning. SPW-Rs have been extensively studied in the adult brain, yet their developmental trajectory is poorly understood. While SPWs have been recorded in rodents shortly after birth, the time point and mechanisms of ripple emergence are still unclear. Here, we combine in vivo electrophysiology with optogenetics and chemogenetics in 4 to 12-day-old mice to address this knowledge gap. We show that ripples are robustly detected and induced by light stimulation of channelrhodopsin-2-transfected CA1 pyramidal neurons only from postnatal day 10 onwards. Leveraging a spiking neural network model, we mechanistically link the maturation of inhibition and ripple emergence. We corroborate these findings by reducing ripple rate upon chemogenetic silencing of CA1 interneurons. Finally, we show that early SPW-Rs elicit a more robust prefrontal cortex response than SPWs lacking ripples. Thus, development of inhibition promotes ripples emergence.

Sharp wave-ripples (SPW-Rs) are the most synchronous events observed in the brain of all mammalian species studied so far. They represent not only a prominent signature of hippocampal electrical activity but serve a critical role in hippocampal function[1,2]. SPW-Rs replay and strengthen recently acquired memories, and facilitate the transfer of information between the hippocampus and other brain regions, most notably the neocortex[3–6].

SPW-Rs consist of two distinct but temporally overlapping events occurring in the CA1 region of the hippocampal formation. The first event, a sharp wave (SPW), is a large-amplitude deflection of the local field potential (LFP) with a negative polarity that emerges in the CA1 radial layer. The second event, a ripple, is a high-frequency oscillation (100–200 Hz) that occurs in the CA1 pyramidal layer[2]. The coordinated activity of large populations of neurons in distinct circuits represents the underlying mechanism of the two events. Excitatory inputs from the CA3 hippocampal area generate large synchronous synaptic currents onto the apical dendrites of CA1 pyramidal neurons (PYRs) that produce the sharp wave[7]. The subtle interplay between local excitatory and inhibitory neuronal populations (INs) is then responsible for generating the rhythmic firing that results in the high-frequency ripple oscillation[8]. The exact roles of excitation and inhibition in ripple generation are still debated, yet the necessity of intact inhibition has been highlighted by several theoretical[9–14] and experimental studies[15–18].

In the hippocampal CA1 region, inhibition is provided by at least 21 distinct types of interneurons (INs)[19,20]. These distinct interneuronal subpopulations display a variety of spiking behaviors during SPW-Rs, ranging from robust firing to complete lack of activity[20]. For example, oriens-lacunosum moleculare cells are suppressed during ripples, while axo-axonic cells fire only at the beginning of the ripple[21]. Conversely, bistratified and trilaminar cells increase their firing rate during SPW-Rs[8,22–24]. The IN-subtype most robustly linked to the generation of ripples are parvalbumin-expressing basket cells (PV⁺-BCs)[8,17,25,26]. These INs fire at high frequency during SPW-Rs, potentially contributing a spike for every ripple cycle, and are phase-locked to the ripple oscillations[7,8].

[1]Institute of Developmental Neurophysiology, Center for Molecular Neurobiology (ZMNH), Hamburg Center of Neuroscience (HCNS), University Medical Center Hamburg-Eppendorf, 20251 Hamburg, Germany. [2]Frankfurt Institute for Advanced Studies, 60438 Frankfurt am Main, Germany. [3]These authors contributed equally: Mattia Chini, Ileana L. Hanganu-Opatz. ✉e-mail: mattia.chini@zmnh.uni-hamburg.de; hangop@zmnh.uni-hamburg.de

Although SPW-Rs have been extensively studied in the adult brain, their development is not well understood. In rodents, SPWs are present as early as postnatal day (P) 3[27], occur bilaterally and longitudinally in a synchronized fashion[28,29], and increase in amplitude along the septo-temporal axis[30]. These early SPWs exhibit unique developmental traits, such as being preceded by body startles[31] and having long "tails" of increased firing before and after their occurrence[27,30]. Differently to their adult counterparts, they rely on incoming excitatory inputs not only from CA3, but also from the entorhinal cortex[32–34].

One of the most prominent features that differentiate these early SPWs from the adult ones is the absence of ripples[27]. However, the developmental stage at which ripples emerge for the first time is subject of debate. In rodents, one study reported that ripples do not emerge until P14[35], whereas another described ripples already from P7 onwards[33]. The late appearance of ripples might be explained by the particularly protracted development of PV+-BCs, one of the IN subpopulations accounting for perisomatic inhibition[32,36]. Moreover, in early development, the excitation-inhibition (E-I) ratio is loose on a temporal scale[37] and shifted towards excitation[32,36,38,39]. However, how changes in inhibition contribute to the development of ripples is still not fully understood.

This knowledge gap is even more striking when considering that a tight coupling between HP and prefrontal cortex (PFC) is present already during early postnatal development, with both theta activity and SPWs in the CA1 area modulating the prefrontal activity[30]. Similar to adult communication[40–44], the CA1 region is the main cortical output of the hippocampus and bidirectionally connected with the PFC. PYRs in CA1 monosynaptically project to the PFC, while the prefrontal feedback involves the midline thalamic nuclei and the entorhinal cortex[45–47]. Whether ripples during development affect prefrontal-hippocampal communication is still unknown.

To address these open questions, we combine in vivo extracellular recordings with optogenetic manipulation of the CA1 area in P4-12 mice. We show that the ripples emerge in the mid of the second postnatal week, concomitantly to a tilting of E-I ratio towards inhibition. Leveraging neural network modeling and chemogenetic manipulation of IN activity, we establish a potential mechanistic link between these two processes. Finally, we report that already at this early stage SPW-Rs elicit a more robust response in the prelimbic subdivision (PL) of the mPFC than SPWs without ripples.

## Results

### CA1 activity scales with age while E-I ratio tilts toward inhibition

To investigate the developmental profile of SPW-Rs, we analyzed a large dataset of extracellular recordings (total duration = 127 h) of local field potential (LFP) and single unit activity (SUA, $n = 1057$ single units) from the hippocampal CA1 area of non-anesthetized P4-P12 mice ($n = 111$ mice) (Fig. 1A and Supplementary Fig. 1A, B, C). To verify the quality and stability of recordings over time, we compared the firing rate in the first and second half of the recording, and quantified the proportion of refractory period violations (RPV), as measure of whether the clustered spikes originate from one unit and, correspondingly, show the characteristic biophysical refractory period (Supplementary Fig. 1D, E). We found that the single unit firing rate of the two halves robustly correlated with each other (0.81 and 0.76, Pearson and Spearman correlation coefficients, respectively) and that the number of RPVs was low (median = 0.34%, below 1% for 92.81% of units).

During the first two postnatal weeks, the hippocampal LFP activity undergoes a massive transition, from an almost entirely isoelectric state, only seldom punctuated by SPWs, to continuous oscillatory activity patterns (Fig. 1B). This developmental change was reflected by the monotonic increase over age of the time fraction with active LFP periods (age slope = 1.01, 95% CI [0.97; 1.05], $p < 10^{-50}$, generalized linear model) (Fig. 1C). The augmented LFP activity was accompanied

by a log-linear rise in SUA firing rate (age slope = 0.27, 95% CI [0.22; 0.33], $p < 10^{-22}$, generalized linear mixed-effect model) (Fig. 1D).

Besides monitoring the main frequency, power, and bandwidth of the periodic component of the power spectrum (PS) of the LFP signal, we analyzed also the aperiodic components of the PS. In particular, the aperiodic component can be expressed by a $1/f^{\chi}$ function, where $f$ is a frequency and $\chi$ is the so-called $1/f$ exponent, which is equivalent to the linear slope of the power spectrum in log-log coordinates. Experimental and theoretical studies have linked the $1/f$ exponent to the network E-I ratio[39,48–53]. In particular, a steeper slope (i.e., a higher $1/f$ exponent) is indicative of an E-I ratio that is more tilted towards inhibition. Since inhibition in CA1 area has been reported to increase during the first two postnatal weeks[32,36], we hypothesized that the $1/f$ exponent should augment too. Indeed, we found that the $1/f$ exponent increased with age and reached constant values around P10 (estimated breakpoint at P10, 95% CI [7.81; 12.19], age slope P4–P10 = 0.16, 95% CI [0.12; 0.21], $p < 10^{-11}$, age slope P10–P12 = −0.26, 95% CI [−0.43; 0.24], $p = 0.14$, piece-wise linear model) (Fig. 1E, F). The developmental change in the $1/f$ exponent was robust across fits in different frequency ranges (Supplementary Fig. 2B), and resembled the dynamics previously reported for the developing mPFC[39].

Thus, similar to other cortical areas, CA1 shows an age-dependent increase of broadband LFP power and SUA firing rate, accompanied by an increase of the $1/f$ exponent indicative of E-I ratio tilting towards inhibition.

### Strengthening the inhibition of PYRs in a neural network model promotes the emergence of ripples

To uncover the emergence mechanisms of developmental ripples and investigate the causal effect of inhibition on their generation, we implemented a spiking neural network representing a simplified circuitry of CA1 (Fig. 2A). As previously published[9] the network architecture comprised an excitatory population (E), representing PYRs, and an inhibitory population (I), representing parvalbumin-positive INs in a ratio set according to anatomical data to 60 to 1[19,54]. Neurons were modeled as leaky integrate-and-fire units, equipped with conductance-based biexponential synapses. Outgoing synapses from PYRs were modeled as AMPA synapses, while INs were implemented with outgoing GABAergic synapses. The network had no spatial structure, and neurons were randomly connected with population-specific probabilities. All units received input noise and in addition to that, to mimic the excitatory input from CA3, a random subpopulation of PYRs received temporally jittered excitatory currents with Gaussian profiles[9,11]. The LFP was calculated as a sum of the absolute values of the AMPA and GABA currents on all excitatory cells[53,55].

In line with previous work[9], when constrained with adult CA1 circuitry parameters, upon arrival of the external drive, the network exhibited ripple-like LFP activity and temporally structured firing in both E and I populations (Fig. 2B). To elucidate the contribution of inhibition to the generation of high-frequency oscillation, we systematically varied the magnitude of the inhibitory conductance between either E-I or I-I populations. In the models with adult I-I conductance and varying levels of E-I inhibition, the external drive consistently increased the firing rate in both E and I populations, but ripple-like activity was not present in all simulations. When E-I inhibition was low (Fig. 2C, left), the firing of the E population did not exhibit a clear temporal structure, and the network did not generate fast-frequency LFP oscillations. When the E-I conductance was raised to levels approaching the adult one (Fig. 2C, right), the LFP generated by the model started exhibiting ripple-like high-frequency oscillations (Fig. 2D). Further increasing the E-I conductance resulted in higher power of the ripple-like oscillations (Fig. 2E, top), but the frequency remained largely stable (median frequency = 150 Hz, Fig. 2E, bottom).

Varying the I-I inhibition while keeping E-I inhibition at adult levels had a distinct effect. In these simulations, upon arrival of the external

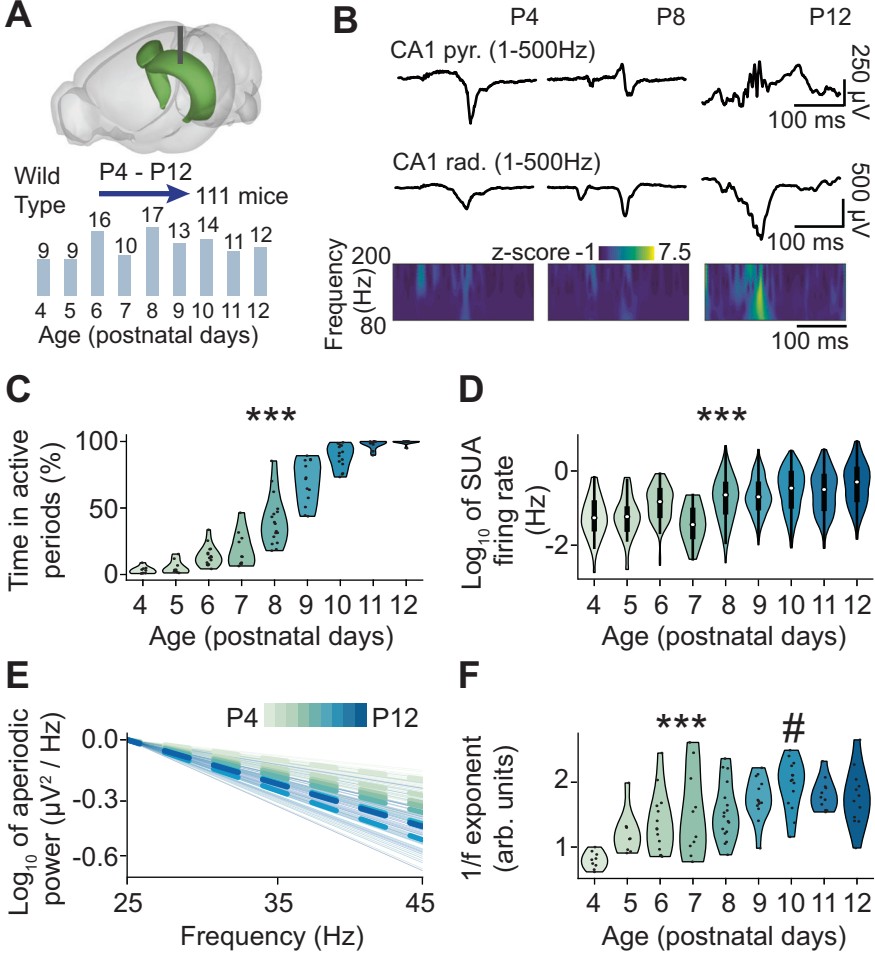

**Fig. 1 | Activity patterns in the hippocampal CA1 area of P4-P12 mice.**
**A** Schematic of the experimental paradigm[82] depicting the location of multi-site electrode in the hippocampal CA1 area (top) and the total number of investigated mice per age group (bottom). **B** Characteristic SPW and SPW-R events extracellularly recorded in the stratum pyramidale and stratum radiale of the hippocampal CA1 area of P4, P8, and P12 mice displayed together with the wavelet spectrum of the LFP at identical timescale. **C** Violin plot displaying the percentage of time spent in active periods of P4–P12 mice ($n = 110$ mice). Generalized (binomial) linear model with logit link function, 95% CI [0.97; 1.05], $p < 10^{-50}$, two-sided. **D** Violin plot with a box plot displaying the single unit (SUA) firing rate of P4–P12 mice (1057 single units from 95 mice). Generalized (gamma) linear mixed-effect model with log link function, mouse as random effect, 95% CI [0.22; 0.33], $p < 10^{-22}$, two-sided. **E** Log–log plot displaying the normalized aperiodic component of the

power spectral density (PSD) in the 25–45 Hz frequency range of P4–P12 mice ($n = 109$ mice). **F** Violin plot displaying the $1/f$ exponent of P4–P12 mice ($n = 109$ mice). Linear model with segmented fit, breakpoint at P10 95% CI [7.81; 12.19], age slope P4–P10 95% CI [0.12; 0.21], $p < 10^{-11}$, age slope P10–P12 95% CI [−0.43; 0.24], $p = 0.14$, two-sided. In (**C**) and (**F**), dots correspond to individual animals. In (**D**) data in the box plot are presented as median (central white circle), interquartile range (thick line) and whiskers (thin lines) extending to the maxima/minima at most 1.5 times the interquartile range. In (**E**) thin lines correspond to individual animals, and thick lines to the mean PSD per age. In (**C**), (**D**), and (**F**) the shaded area represents the probability distribution density of the variable. In (**C**), (**D**), and (**F**), asterisks indicate a significant effect of age. ***$p < 0.001$. Hash (#) indicates estimated breakpoints in the linear piece-wise regression. Source data are provided as a Source Data file.

stimulus, the network always exhibited high-frequency oscillations, regardless of the strength of the I-I inhibition (median frequency = 140 Hz) (Fig. 2F, G). However, increasing the strength of the I-I inhibition increased the power (Fig. 2H, top) and the frequency (from 100 to 180 Hz, Fig. 2H, bottom) of the oscillations.

Thus, simulations of the CA1 local circuitry model reveal that high levels of E-I inhibition are needed for the emergence of ripple-like activity, while the strength of I-I inhibition mainly modulates its frequency.

**Ripples emerge halfway through the second postnatal week**
The in silico data above are instrumental for subsequent experimental approaches, since they predict that an increase in inhibition is necessary for the generation of ripple-like LFP activity. Since from P4 to P10 we observed an increase of the $1/f$ exponent as an indicator of a shift of E-I ratio towards inhibition, we hypothesized that ripples emerge

during this developmental phase. To test this hypothesis, we separately evaluated the age-dependent presence of ripples in the frequency and time domain.

In the fast frequency band (80–200 Hz), we found that the average power exponentially increased over age (age slope = 0.24, 95% C.I. [0.19; 0.28], $p < 10^{-14}$, generalized linear model) (Fig. 3A, B). However, an increase of power in a certain frequency band could simply reflect a broadband power increase (the offset of the aperiodic component). To disentangle the aperiodic and periodic contributions to the power increase, we used FOOOF, a method that separately parametrizes the two components[49] (Fig. 3C, D, and Supplementary Fig. 3A). Using this approach, we found that both components contributed to the power increase in the ripple frequency band. On the one hand, we identified a broadband exponential increase of the PS offset (age slope = 0.25, 95% C.I. [0.19; 0.30], $p < 10^{-12}$, generalized linear model) (Fig. 3E). On the other hand, from

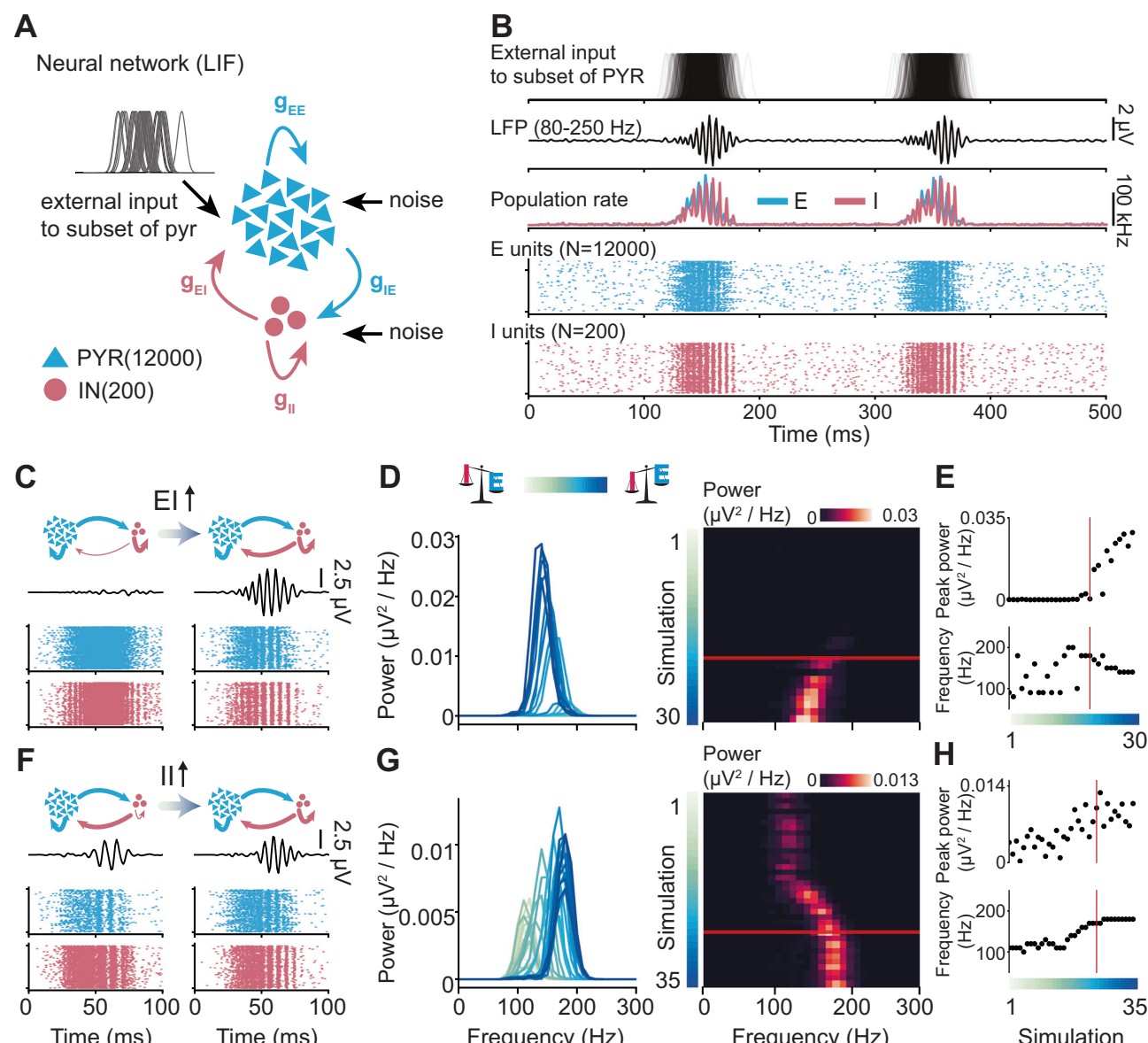

**Fig. 2 | Effect of increasing inhibition on ripples in a neural network model.**
**A** Schematic of the spiking neural network model. **B** Activity patterns simulated in the network with "adult"-levels of I-to-E and I-to-I inhibition. **C** Reconstructed LFP in 80–250 Hz frequency range (top) and raster plots (bottom) displaying units firing in response to the external input. Left: network with a low level of I-to-E inhibition. Right: network with an "adult" level of I-to-E inhibition. **D** Mean power spectral density for 30 simulations with increasing I-to-E inhibition. **E** Change in peak power (top) and frequency (bottom) over increasing levels of inhibition. **F**, **G**, **H** are same as (**C**), (**D**), (**E**) for I-to-I inhibition. In (**D**), (**E**), (**G**), and (**H**), the red line marks an "adult"-like level of inhibition. In (**D**), (**E**), (**G**), and (**H**), simulation 1 is the network with the lowest level of inhibition, and simulation 30/35 is the one with the highest. Source data are provided as a Source Data file.

P9 onwards, FOOOF detected peaks in the PS that are indicative of bona fide oscillatory phenomena in the ripple frequency band (Fig. 3F). The power of these peaks slightly increased over age (Fig. 3G, left) (age slope = 0.01, 95% CI [−0.001; 0.02], $p = 0.078$, linear model), whereas their central frequency (Fig. 3G, right) and width (Supplementary Fig. 3B) showed no age-related changes (age slope = 2.14, 95% CI [−5.35; 9.62], $p = 0.55$ and age slope = −0.34, 95% CI [−4.88; 4.20], $p = 0.88$, respectively, linear model).

For ripple detection in the time domain (Fig. 4A, B), we separately detected SPWs and ripples. Ripples were considered for further analysis only if they co-occurred with SPWs. In line with previous studies[5,30,33,35,46], SPWs were detected by thresholding the LFP signal in the CA1 radial layer (Fig. 4A, B), with a threshold ranging from 3 to 5 standard deviations. As the mean and variance of the LFP signal in our dataset exponentially increased over age, we used an adaptive threshold multiplier comprised between 3 and 5 standard deviations and inversely proportional to the signal variance (Supplementary Fig. 4C). The detection of ripples in the time domain relies on heuristically set thresholds and post hoc manual curation, a process that is inherently prone to errors, as recently highlighted[56]. To avoid biases, we used two different methods for the detection of ripples in the time domain (Fig. 4A, B, Supplementary Fig. 4A). The first method is based on thresholding the power of the LFP signal after narrow-band filtering in the frequency band of interest. This approach is commonly used to detect ripples in adult animals[5,17]. The second method[57] works without narrow-band filtering. By-cycle relies on splitting the LFP signal into putative cycles and, based on cycle features, on determining whether individual cycles are part of a genuine oscillatory event. Only events identified by both detection methods were considered as ripples in the present study.

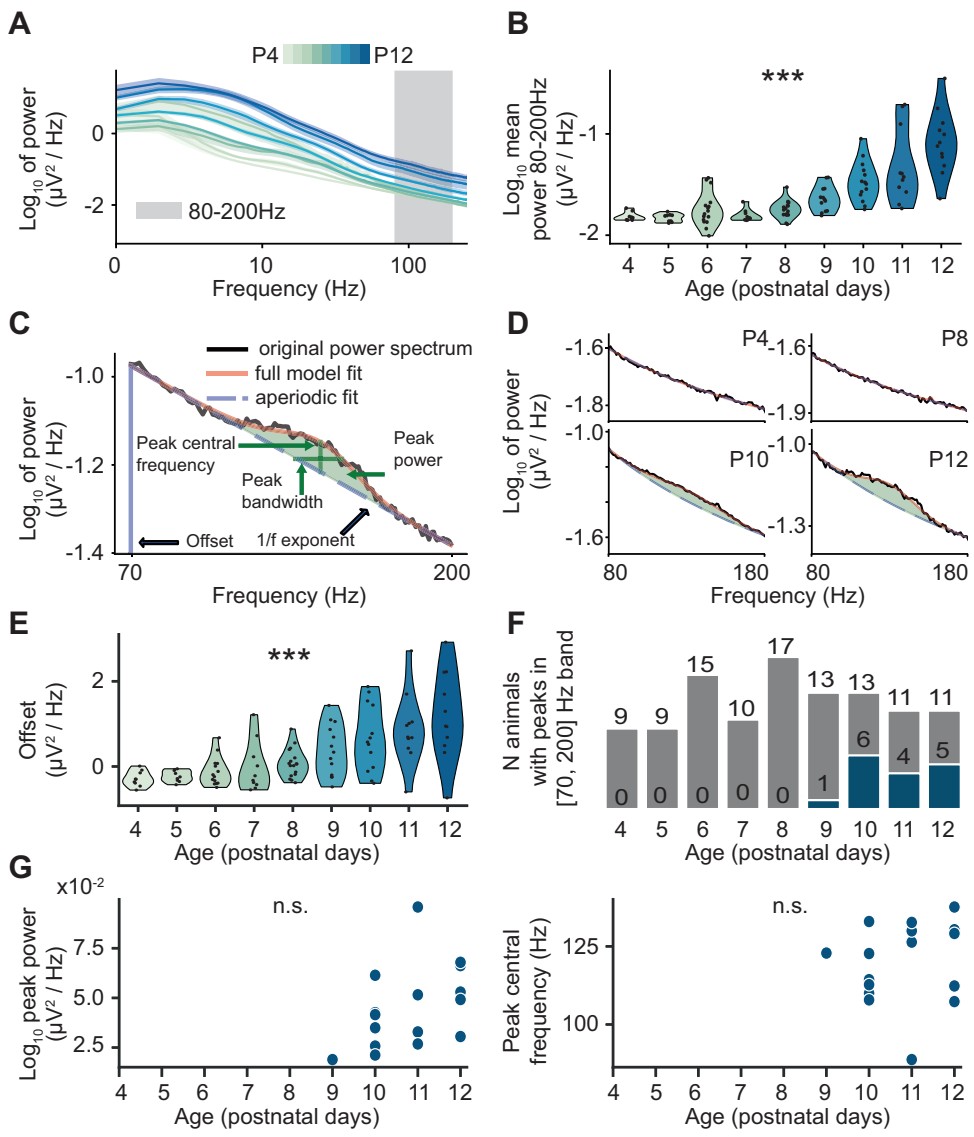

**Fig. 3 | Developmental profile of fast oscillatory activity in the CA1 area of P4–P12 mice. A** Power spectral density (PSD) of hippocampal activity of P4–P12 mice (mean ± s.e.m., $n = 111$ mice). **B** Violin plot displaying the power in the 80-200 Hz frequency band across age ($n = 111$ mice). Generalized (gamma) linear model with log link function, 95% C.I. [0.19; 0.28], $p < 10^{-14}$, two-sided. **C** Example of a parameterized power spectrum in the 70–200 Hz frequency range from a P12 mouse. **D** Representative examples of parameterized power spectra in the 70–200 Hz frequency range recorded from P4, P8, P10, and P12 mice. **E** Violin plot displaying the power spectrum offset across age ($n = 108$ mice). Generalized (gamma) linear model with log link function, 95% C.I. [0.19; 0.30], $p < 10^{-12}$, two-

sided. **F** Distribution of high-frequency peaks over age of P4-P12 mice ($n = 108$ mice). Gray bars indicate the total number of mice per age group, while blue bars indicate the number of mice with detected peaks. **G** Characteristics of detected fast frequency peaks. Left, log peak power, and right, peak central frequency. Linear model, left 95% CI [−0.001; 0.02], $p = 0.078$, right 95% CI [−5.35; 9.62], $p = 0.55$, two-sided. In (**B**), (**E**), and (**G**), dots correspond to individual animals. In (**B**) and (**E**), the shaded area represents the probability distribution density of the variable. In (**B**) and (**E**), asterisks indicate a significant effect of age. ***$p < 0.001$. Source data are provided as a Source Data file.

As previously reported[27], SPWs were present already at P4, and their rate exhibited a nonmonotonic change over age (Fig. 4C). Both the SPW-R rate and the percentage of SPWs with ripples showed no change from P4 to P10, followed by an increase from P10 to P12 (estimated break-point 9.75 95% C.I. [8.69; 10.82]; P4–P10 age slope = −0.0028, 95% C.I. [−0.17; 0.16], $p = 0.974$, P10–P12 age slope = 0.80, 95% C.I. [0.36; 1.23], $p = 0.00096$, piece-wise linear model) (Fig. 4D). (estimated break-point 8.97 95% C.I. [7.41; 10.53]; P4–P9 age slope = −0.0051, 95% C.I. [−0.02; 0.01], $p = 0.47$, P9-P12 age slope = 0.03, 95% C.I. [0.01; 0.05], $p = 0.0034$, piece-wise linear model) (Fig. 4E). The SPW-R rate during the investigated developmental window did not change for different frequency bands used for ripple detection (Supplementary Fig. 5A–C, Supplementary Table S5). Further

corroborating the robustness of the detection approach, the presence of an oscillatory peak in the power spectrum (detected in the frequency domain) significantly correlated with the SPW-R rate (detected in the time domain) for all frequency bands (Supplementary Fig. 5D, Supplementary Table S5).

To control for the possible effect of body movement on SPW-R rate, we recorded a separate group of P11–12 mice ($n = 7$ mice) while simultaneously video-monitoring their motor activity. We detected no differences in SPW-R rate between periods with and without movement (condition effect = −0.13, 95% CI [−1.49; 1.23], $p = 0.837$) (Supplementary Fig. 6).

Even though the SPW-R rate conspicuously increased starting from P10, few SPW-Rs were detected also at earlier ages. However, the

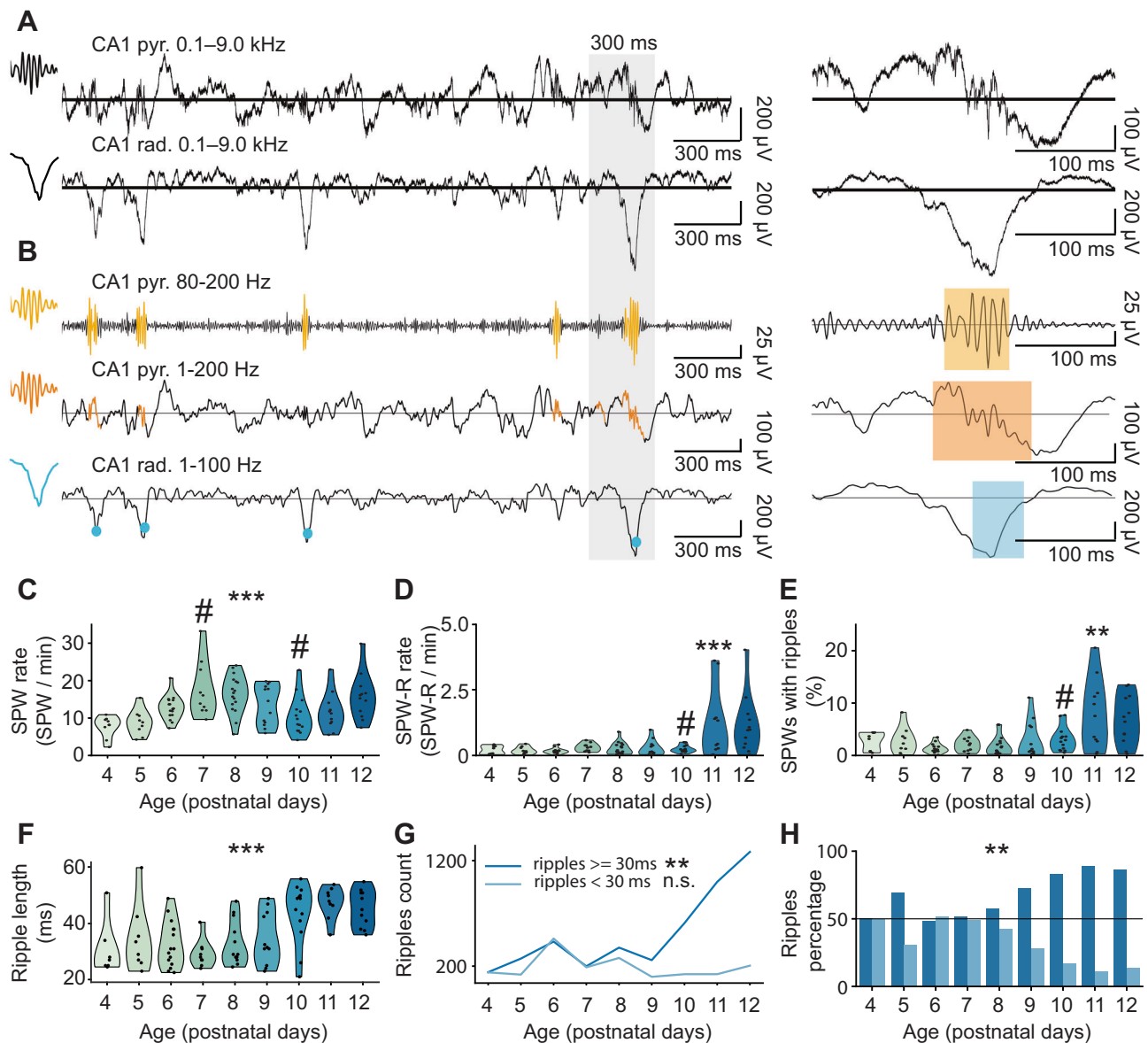

**Fig. 4 | Developmental emergence of SPWs and ripples in the CA1 area of P4–P12 mice. A** A characteristic example of 3s long LFP traces recorded in the stratum pyramidale and stratum radiale of the hippocampal CA1 area of a P12 mouse (left), and zoomed-in 300 ms long LFP traces of one SPW-R event (right). **B** LFP traces shown in (A) highlighting the detected ripples (top and middle) and SPW (bottom). **C** Violin plot displaying the SPWs rate of P4-P12 mice (n = 111 mice). Linear model with segmented fit, breakpoints at 7.21 95% C.I. [6.40; 8.00] and 10.56 95% C.I. [9.94; 11.19]; P4–P7 age slope 95% C.I. [2.23; 5.86], p < 10⁻⁴, P7–P10 age slope 95% C.I. [−5.47; −0.77], p < 10⁻⁵, P10–P12 age slope 95% C.I. [2.97; 13.51], p = 0.00017, two-sided. **D** Violin plot displaying the SPW-R rate of P4–P12 mice (n = 111 mice). Linear model with segmented fit, breakpoint at 9.75 95% C.I. [8.69; 10.82]; P4–P10 age slope 95% C.I. [−0.17; 0.16], p = 0.974, P10-P12 age slope 95% C.I. [0.36; 1.23], p = 0.00096, two-sided. **E** Violin plot displaying the percentage of SPWs with ripples of P4–P12 mice (n = 111 mice). Linear model with segmented fit, breakpoint at

8.97 95% C.I. [7.41; 10.53]; P4–P9 slope 95% C.I. [−0.02; 0.01], p = 0.47, P9–P12 age slope 95% C.I. [0.01; 0.05], p = 0.0034, two-sided. **F** Violin plot showing ripple length over age (n = 111 mice). Linear mixed-effects model, mouse as random effect, 95% C.I. [9.33; 15.74], p < 0.0001, two-sided, Tukey correction for multiple comparisons. **G** Line plot displaying the absolute number of short and long ripples over age. Linear model, long ripples age slope 95% CI [48.93; 189.37], p = 0.00511, short ripples age slope 95% CI [−45.15; 28.55], p = 0. 6108, two-sided. **H** Bar plot showing the percentage of long and short ripples at each age. Linear model, long ripples percentage age slope 95% CI [1.87; 7.87], p = 0.0064, short ripples percentage age slope 95% CI [−7.87; −1.87], p = 0.0064, two-sided. In (C), (D), (E), and (F), the shaded area represents the probability distribution density of the variable, dots correspond to individual animals, and asterisks indicate a significant effect of age. **p < 0.01, ***p < 0.001. Hash (#) indicates estimated breakpoints in linear piece-wise regression. Source data are provided as a Source Data file.

average ripple length in the early ages was significantly shorter compared to older ages (median ripple length in P4–P9 32 ms, median ripple length in P10-P12 47 ms, effect of age on ripple length (P4–P9 vs P10–P12) = 12.53 ms, 95% C.I. [9.33; 15.74], p < 0.0001) (Fig. 4F). The length of 30 ms roughly corresponds to maximum three oscillatory cycles at >100 Hz. It is therefore questionable whether such few cycles are sufficient to represent a genuine oscillatory event. In addition, it is possible that population synchrony in the form of short non-rhythmic

spike bursts occurring upon the arrival of SPW might have been detected as short ripples due to filtering effects[56,58]. The number and the proportion of longer ripples, with a duration comparable to those observed in adult animals (3–9 cycles, according to[2]), markedly increased after P10 (long ripples age slope = 119.2, 95% CI [48.93; 189.37], p = 0.00511, short ripples age slope = −8.30, 95% CI [−45.15; 28.55], p = 0. 6108, linear model) (Fig. 4G). (long ripples percentage age slope = 4.874, 95% CI [1.87; 7.87], p = 0.0064, short ripples

percentage age slope = −4.874, 95% CI [−7.87; −1.87], $p$ = 0.0064, linear model) (Fig. 4H). (Supplementary Fig. 7A, Supplementary Table S5). Similar to data from the adult HP[58], we also detected "solo" ripples (i.e., ripples that did not co-occur with SPWs) in all age groups (Supplementary Fig. 7B), yet these events were excluded from further analysis.

These data indicate that ripples are detectable in the frequency and time domain from P10 on, when the value of the $1/f$ exponent stabilizes. Their power but not frequency augments with age, in line with the results of neural network simulations.

### Developmental changes in the temporal structure of single unit activity during SPWs

Since the interplay between excitatory and inhibitory neurons appears to be critical for ripple generation, we investigated how developmental changes in the temporal structure of neuronal firing underpin the emergence of ripples.

In agreement with previously published data[27,34,35], spiking activity was positively modulated during ~90 % of all SPWs (Fig. 5A). The firing increase was most prominent during SPW-Rs than during SPWs lacking ripples (Fig. 5B). The firing rate during both baseline and SPWs/SPW-Rs was higher in older animals (firing rate during SPW age slope = 0.09, 95% CI [0.06; 0.12], $p < 10^{-8}$, firing rate during SPW-R age slope = 0.14, 95% CI [0.05; 0.22], $p$ = 0.0014, linear model), yet the relative firing increase was similar in all ages, being at ~3 standard deviations (Fig. 5C, top). In addition, the number of units contributing to SPW events, as well as the reliability of their participation, increased with age (age slope = 0.414, 95% CI [0.296; 0.531], $p < 10^{-9}$, linear model) (Supplementary Fig. 8A) (age slope = 1.13, 95% CI [0.603; 1.658], $p < 10^{-4}$, linear mixed-effect model) (Supplementary Fig. 8B).

The comparable firing increase from P4 to P12 might indicate that the age-dependent emergence of ripples relates to the fine temporal structure of single-unit activity during SPWs. To test this hypothesis, we quantified the timing of the population firing rate peak, the offset of the first spike, and the number of spikes that individual units fired during SPWs. The timing of the population firing peak varied with age, peaking after the SPW maximum amplitude in younger animals and before the SPW maximum amplitude in older mice (Fig. 5C, top). To test whether the different timing is the result of the increasing inhibitory tone, we quantified the population activity peak as we varied the inhibition strength in neural network simulations. In line with this hypothesis, the dynamics in the model network with increasing E-I conductance mirrored the experimentally observed dynamics (Fig. 5C, middle). In the network with increasing I-I conductance, no difference in the peak was observed (Fig. 5C, bottom). The shift in firing rate peak timing with age indicate a stronger inhibition exerted onto PYRs.

A wealth of studies documented that PYRs and INs exhibit a distinct temporal firing patterns during SPW-Rs in adult rodents. PYRs typically fire once per event, and the timing of their individual firing is determined by the magnitude of the pre-SPW hyperpolarization they receive, with larger hyperpolarizations resulting in later spiking[59]. INs have much higher firing rates, and can contribute with up to a spike to every ripple cycle[8,60]. We, therefore hypothesized that the developmental increase of IN involvement in ripples might result in more temporarily jittered PYR firing and a higher number of units firing multiple times per SPW event. In line with this hypothesis, at younger ages, the offset of the first spike had a very narrow distribution, as most units fired close to the maximum amplitude of the SPW. In older animals, the variance of the distribution was higher, with more units first engaged at earlier or later time points in the SPW (Fig. 5D, top). Also conforming to the hypothesis of a greater contribution of INs in older mice, the percentage of units firing several spikes per SPW event increased with age (Fig. 5D, middle). This effect was particularly strong (i.e., two-fold increase) in cells contributing with four or more spikes per SPW event (Fig. 5D, bottom).

Thus, while during SPWs the neuronal firing increased in all age groups, subtle age-dependent changes in the firing temporal structure reflect a greater contribution of INs to the developmental emergence of ripples.

### Optogenetic activation of CA1 pyramidal neurons induces high-frequency oscillations from P10 on

In adult rodents, optogenetic activation of PYRs in CA1 area with a rectangular light stimulus produces ripple-like induced high-frequency oscillations (iHFO)[5,17,60]. This kind of stimulus is considered to mimic the incoming wave of excitation that takes place during SPWs[5,17]. To confirm that ripples emerge around P10, we therefore tested whether an analogous optogenetic approach induced iHFOs in P8-12 mice.

To this aim, we used a previously developed manipulation protocol[30] and specifically transfected CA1 PYRs with a construct carrying the fast and efficient channelrhodopsin-2 mutant (ChR2E123T/T159C) and the red fluorescent protein tDimer2 by in utero electroporation (IUE). An opsin-free control group underwent the same IUE protocol, with a construct carrying only tDimer2. This IUE protocol targets PYRs in the CA1 area of both dorsal HP (dHP) and intermediate/ventral HP (i/vHP) and does not alter the development at embryonic and postnatal stages[30,46,47]. For these experiments, we pooled data from recordings performed in either the dorsal or ventral/intermediate CA1, depending on the expression localization (Fig. 6A). The number of transfected cells was similar across all ages (Fig. 6A) (age slope = −3.61, 95% C.I. [−10.13; 2.91], $p$ = 0.261, linear model).

Blue light tapered-in pulse stimulation (473 nm, 2.4−54 mW at fiber tip, 30 sweeps, 120 ms-long pulses) at all light intensities augmented SUA in all investigated mice (Fig. 6B, C, and Supplementary Fig. 9A). For P8 and P9 mice, augmenting laser power evoked a stronger and faster spiking response, yet failed to induce iHFOs. Only from P10 onwards, light stimulation led to the generation of ripple-like iHFOs (age slope = 0.73, 95% CI [0.67; 0.81], $p < 10^{-50}$, generalized linear model) (Fig. 6D and Supplementary Fig. 9B). At this age, increasing the light intensity augmented the reliability of evoking iHFOs (Supplementary Fig. 9C). Light-evoked iHFO were accompanied by prominent spiking with features highly reminiscent of those observed during recorded SPW-Rs. In particular, P10−P12 mice exhibited a wider distribution of the first spike offset (Fig. 6E, top) and a twofold increase in units spiking four or more spikes per single stimulation period (Fig. 6E, bottom).

These data confirm that, only from P10 onwards, optogenetic stimulation of PYRs in the CA1 area induces ripple-like iHFOs and spiking signatures similar to those that have been described in adult mice. Conversely, at younger age, the optogenetic stimulation only increased the firing activity in a temporally unstructured manner.

### Chemogenetic inhibition of CA1 interneurons reduces ripple rate in P11−P12 mice

To strengthen the in silico evidence highlighting the role of inhibition in ripple generation with a mechanistic experimental approach, we chemogenetically silenced INs in the CA1 area of P11−12 mice. We hypothesized that if inhibition plays a role in generating ripples, this manipulation should decrease their amount.

To this aim, we transfected Dlx5/6[cre] INs with inhibitory DREADDs by injecting the construct hM4D(Gi) (AAV9-EF1a-DIO-hM4D(Gi)-mCherry) into the CA1 of P0-P2 Dlx5/6[cre] mice. Subsequently, we recorded the CA1 activity in vivo before (45 min, baseline) and after administration (45 min) of C21 (3 mg/kg, i.p.), a synthetic activator of DREADDs[61] (Fig. 7A). We controlled for potential non-specific effects of C21 administration by including a group of controls comprising Dlx5/6[cre]-negative mice, Dlx5/6[cre]-positive mice that did not express the virus, and wild type mice.

In line with our hypothesis, C21-treated Dlx5/6[cre]-positive mice exhibited a considerable reduction in SPW-R rate compared to the control group (condition effect = −0.56, 95% CI [−1.01; −0.12],

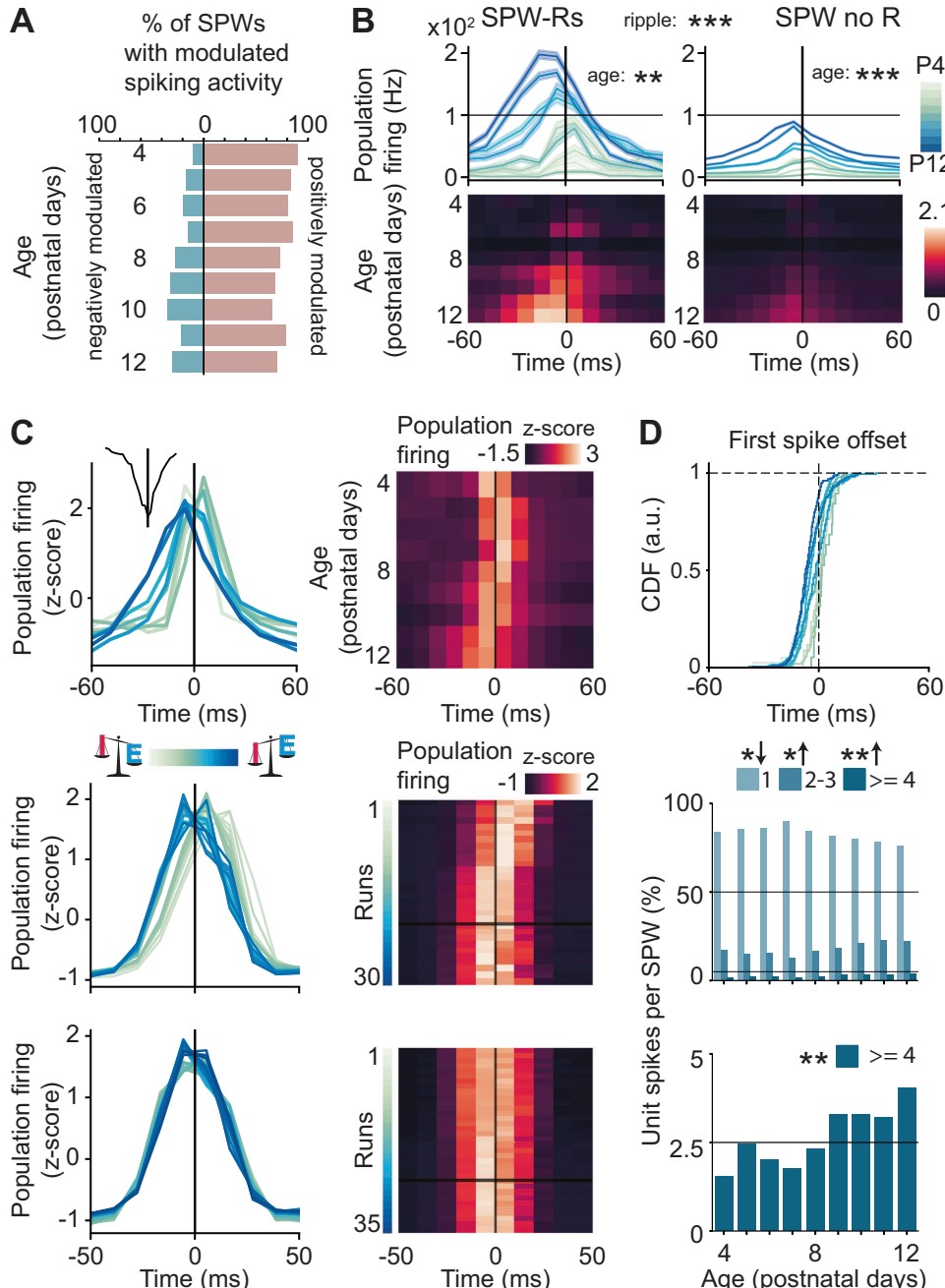

**Fig. 5 | Single unit activity during SPWs/SPW-Rs in the hippocampal CA1 area of P4–P12 mice. A** Percentage of SPWs with modulated population spiking activity (*n* = 111 mice). **B** Population firing during SPW-Rs and SPWs shown as mean ± s.e.m (line plots, top) and mean (heatmaps, bottom) (*n* = 111 mice). Linear model, firing rate during SPW age slope 95% CI [0.06; 0.12], $p < 10^{-8}$, firing rate during SPW-R age slope 95% CI [0.05; 0.22], *p* = 0.0014; Linear mixed-effects model, mouse as random effect, condition (presence or absence of a ripple), *p* < 0.0001, two-sided. **C** Line plot (left) and heatmap (right) of the population firing rate with respect to the timing of SPW as a function of age in experimental data (top), and as a function of inhibition strength with respect to the timing of external drive in the model with

increasing I-to-E inhibition (middle) and in the model with increasing I-to-I inhibition (bottom). On the heatmaps, the horizontal black line marks an "adult"-like level of inhibition. **D** Line plot displaying the cumulative distribution function of the first spike offset during the SPW (top), and bar plots displaying the percentage of units contributing 1, 2–3, or 4 or more spikes to the SPW (middle and bottom). Linear model, 1 spike age slope *p* = 0.0176, 2-3 spikes age slope *p* = 0.01014, >=4 spikes age slope *p* = 0.00148, two-sided. In (**D**), arrow down indicates a negative effect of age, whereas arrow up corresponds to a positive effect of age. In (**B**) and (**D**), asterisks indicate a significant effect of age or ripple. * *p* < 0.05, ** *p* < 0.01, *** *p* < 0.001. Source data are provided as a Source Data file.

*p* = 0.017, linear model) (Fig. 7B, C). The reduction in detected SPW-Rs took place despite an increase in SUA firing rate after C21 administration (condition effect = 0.38, 95% CI [0.05; 0.71], *p* = 0.042, linear mixed-effect model) (Fig. 7D–F). This data strengthens the evidence that inhibition is necessary for ripple generation and that this process relies on the balanced interplay between INs and PYRs, and not merely on increased firing rate.

## Ripples strengthen the coupling between mPFC and HP during SPWs

Hippocampal SPW-Rs have been shown to drive a robust increase of activity in the mPFC at adult age[62–64]. Similarly, a robust hippocampal-prefrontal coupling is already present towards the end of the first postnatal week in rodents[30,46], yet it is still unknown how the emergence of ripples impacts the developing prefrontal activity. To fill this

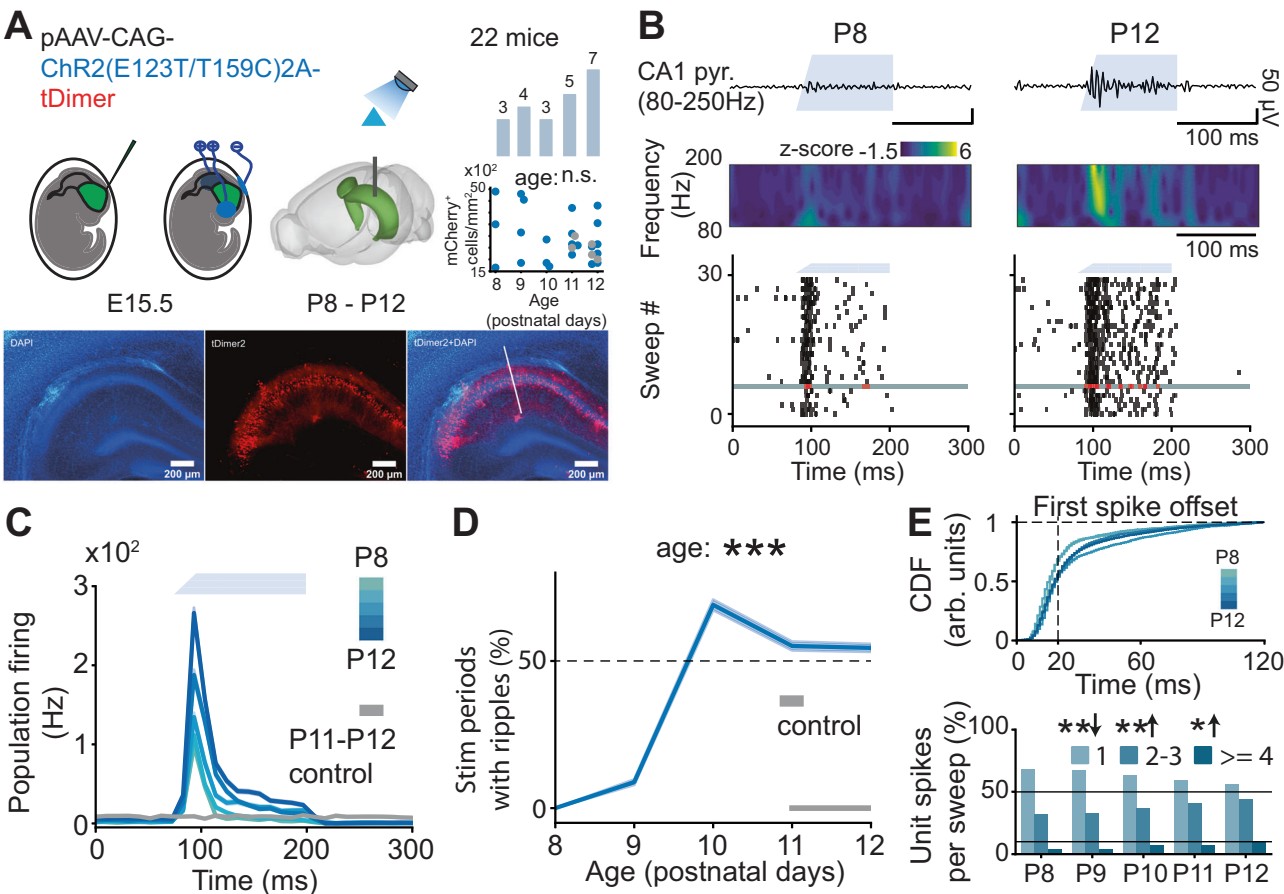

**Fig. 6 | Optogenetic activation of CA1 pyramidal neurons modulates activity in the CA1 area of P8-P12 mice. A** Schematic of the experimental protocol for optogenetic activation of ChR2-transfected CA1 pyramidal neurons, the total number of investigated mice per age group and the quantification of transfected cells across age (top) and tDimer2- and ChR2-expressing cells (red) in the CA1 of a P11 mouse (bottom). The white line marks the position of the recording electrode (*n* = 22 mice). **B** Characteristic example of optogenetically evoked LFP activity extracellularly recorded in the stratum pyramidale of the hippocampal CA1 area of P8 and P12 mice displayed together with wavelet spectrum of LFP at identical time scale (top), and raster plots displaying single unit firing in response to 30 sweeps of optogenetic stimulation (bottom). The LFP plot corresponds to the sweep highlighted in gray. **C** Population firing rates in response to the optogenetic stimulation of P8–P12 mice (*n* = 22 mice). **D** Percentage of stimulation periods with iHFO of P8-P12 mice (*n* = 22 mice). Generalized (binomial) linear model with logit link function, 95% CI [0.67; 0.81], *p* < 10$^{-50}$, two-sided. **E** Cumulative distribution function of the first spike offset in response to the optogenetic stimulation (top) and percentage of units contributing 1, 2–3, or 4 or more spikes to the population activity during stimulation (bottom) (*n* = 22 mice). Linear model, 1 spike age slope *p* = 0.00198, 2–3 spikes age slope *p* = 0.00198, >=4 spikes age slope *p* = 0.0129, two-sided. In (**C**) and (**D**), data are shown as mean ± s.e.m. In (**E**), the arrow down indicates a negative effect of age, whereas the arrow up corresponds to a positive effect of age. In (**D**) and (**E**), an asterisk indicates a significant effect of age. *\*p* < 0.05, *\*\*p* < 0.01, *\*\*\*p* < 0.001. Source data are provided as a Source Data file.

knowledge gap, we analyzed SUA (*n* = 2457 single units from 85 mice) simultaneously recorded from the prelimbic subdivision (PL) of the mPFC and the hippocampal CA1 area of non-anesthetized P4-P12 mice (Fig. 8A).

During the entire developmental time window, the prelimbic firing sharply increased during SPWs, yet no age-dependent differences were detected (firing rate during SPW age slope = 0.026, 95% CI [−0.005; 0.058], *p* = 0.108, firing rate during SPW-R age slope = 0.009, 95% CI [−0.14; 0.16], *p* = 0.909, linear model) (Fig. 8B). The firing increase in PL was larger during SPW-Rs than during SPWs without ripples (Fig. 8B, right), indicating that ripples tighten the hippocampal-prefrontal coupling.

In addition, we evaluated the effects of light stimulation of hippocampal CA1 on the prelimbic spiking (*n* = 696 single units from 19 mice) (Fig. 8C, D). Despite the rather low number of PYRs that are transfected with IUE[30], and the fact that not only the i/vHP strongly projecting to PL but also the dHP with weak connectivity is stimulated, light pulses (473 nm, 2.4–54 mW at fiber tip, 30 sweeps, 120 ms-long) evoked spiking activity in PL (Fig. 8D, E). The effect increased with age and at P12, the oldest investigated age, a

prominent spiking peak was detected at 100–150 ms onset from light stimulus.

The data show that both ripples and optogenetically-induced ripple-like activity strengthen the developmental drive from hippocampal CA1 area to PL.

## Discussion

SPW-Rs are the most synchronous pattern of electrical activity occurring in the mammalian brain. Despite being one of the most abundantly researched phenomena in the adult brain, relatively little is known about their developmental emergence and the underlying mechanism. Shortly after birth, SPWs have several distinctive features, such as being preceded by body startles[31], being accompanied by increased firing before and after their occurrence[27,30], and not co-occurring with ripples. Here, we showed that both spontaneous ripples and optogenetically-induced ripple-like activity emerge around P10 in mice. At the same time, the E-I ratio in the CA1 network tilts towards inhibition. By leveraging neural network modeling and chemogenetic manipulation of IN activity, we established a potential mechanistic link between the rise of inhibition and the emergence of ripples. We

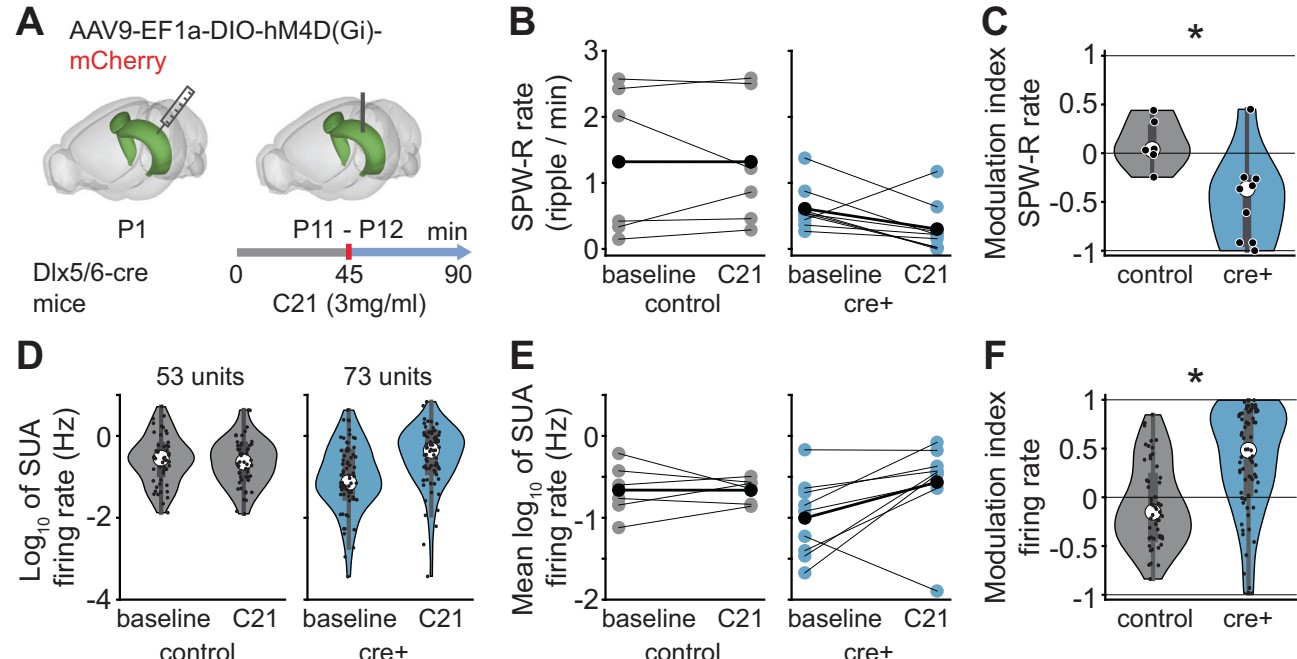

**Fig. 7 | Effects of silencing interneurons by inhibitory DREADDs on SPW-Rs in P11-P12 mice. A** Schematic of the experimental protocol for chemogenetic silencing of Dlx5/6cre+ interneurons. **B** Scatterplot displaying the SPW-R rate for control (left, $n = 6$) and Dlx5/6cre+ (right, $n = 9$) mice before and after C21 injection. **C** Violin plot with a box plot displaying the C21 injection modulation index of SPW-R rate in control ($n = 6$) and Dlx5/6cre+ ($n = 9$) mice. Black dots correspond to individual animals. Linear model, condition effect 95% CI [−1.01; −0.12], $p = 0.017$, two-sided, Tukey correction for multiple comparisons. **D** Violin plot with a box plot displaying the single unit firing rate in control (left, 53 single units from 6 mice) and Dlx5/6cre+ (right, 73 single units from 9 mice) mice before and after C21 injection. Black dots correspond to individual single units. **E** Scatterplot displaying the mean single unit firing rate for control (left, $n = 6$) and Dlx5/6cre+ (right, $n = 9$) mice before and after

C21 injection. **F** Violin plot with a box plot displaying the C21 injection modulation index of single unit firing rate in control (53 single units from 6 mice) and Dlx5/6cre+ (73 single units from 9 mice) mice. Linear mixed-effect model, mouse as random effect, condition effect 95% CI [0.05; 0.71], $p = 0.042$, two-sided Tukey correction for multiple comparisons. In (**B**) and (**E**), blue/gray dots on the scatterplot correspond to individual animals while black dots represent mean values. In (**C**), (**D**), and (**F**), data in the box plot are presented as median (central white circle), interquartile range (thick line) and whiskers (thin lines) extending to the maxima/minima at most 1.5 times the interquartile range. The shaded area on the violin plot represents the probability density distribution of the variable. In (**C**) and (**F**), asterisks indicate a significant effect of condition. *$p < 0.05$. Source data are provided as a Source Data file.

corroborate these findings by showing that the fine temporal structure of spiking during SPWs displays age-related changes that are suggestive of inhibition strengthening. Finally, we explore the coupling between mPFC PL and CA1 and show that SPWs co-occurring with ripples have a larger impact on PL than those without.

Despite substantial interest in SPW-Rs, ripple detection algorithms often rely on arbitrary thresholds and post hoc manual verification, with little consistency across studies[56]. In neonatal animals, the detection is further complicated by the low amplitude and discontinuous nature of the LFP activity, as well as uncertainty about the developmental stage at which ripples firstly appear. It is therefore not surprising that contradictory results regarding the development of SPW-Rs have been reported. While coarse inspection of ripple frequency band led to the conclusion that ripples do not emerge before P14 in rats[35], a detection approach in the time domain argued that these fast events are present already at P7[33]. To reconcile these discrepancies, we employed a multi-layered approach. Firstly, we parameterized the frequency domain into aperiodic and periodic components. We identified a log-linear increase in the power of the ripple frequency band in mice older than P4. However, power spectra peaks, indicative of genuine oscillatory phenomena, were detected only from P10 onwards. Subsequently, we detected ripples in the time domain, using a rigorous approach that considered as such only events co-occurring with SPWs and identified by two independent algorithms. In agreement with the analysis in the frequency domain, we only detected ripples of substantial length from P10 onwards. Lastly, the developmental time point of ripple emergence was confirmed by optogenetic manipulation. Light stimulation of ChR2-transfected CA1

PYRs induced ripple-like fast-frequency oscillations only from P10 on. Thus, we present converging evidence that highlights the mid of the second postnatal week as the developmental phase in which ripples first appear.

Although the mechanisms underlying ripple generation in the adult brain remain a topic of debate, a growing body of evidence suggests that inhibition plays a crucial role. Three classes of in silico ripple-generation models have been previously proposed[2,9,17]. In the first class, "interneuron ripples", the oscillation is generated by reciprocally connected fast-spiking INs receiving an excitatory input. The rhythmicity is inherited by PYRs via paced inhibition. In the second class of models, "pyramidal ripples", the oscillation is generated by PYRs coupled via gap junctions, irrespective of INs. In the third class, "pyramidal-interneuron ripples", the oscillation is generated by recurrently coupled PYRs and INs, with PYRs being the receivers of the external drive. This model is the most compatible with in vivo experimental data[17]. Indeed, while optogenetically stimulating CA1 PYRs induces ripple-like activity, doing so in PV+ INs failed to induce ripples, which is inconsistent with the "interneuron ripple" model. Additionally, local blocking of GABA_A receptors with picrotoxin has been shown to suppress high frequency oscillations[17]. These data showed that PYRs alone also fail to generate ripple and contradicted the "pyramidal ripples" model. Further, during SPW-Rs, inhibition dominates excitation with a peak conductance ratio of $4.1 \pm 0.5$[15]. Taken together, these results suggest that the interaction between PYRs and INs in the adult CA1 area is necessary to generate ripples. Similarly, we showed that during early development, chemogenetically silencing interneurons interferes with ripple generation and results in a

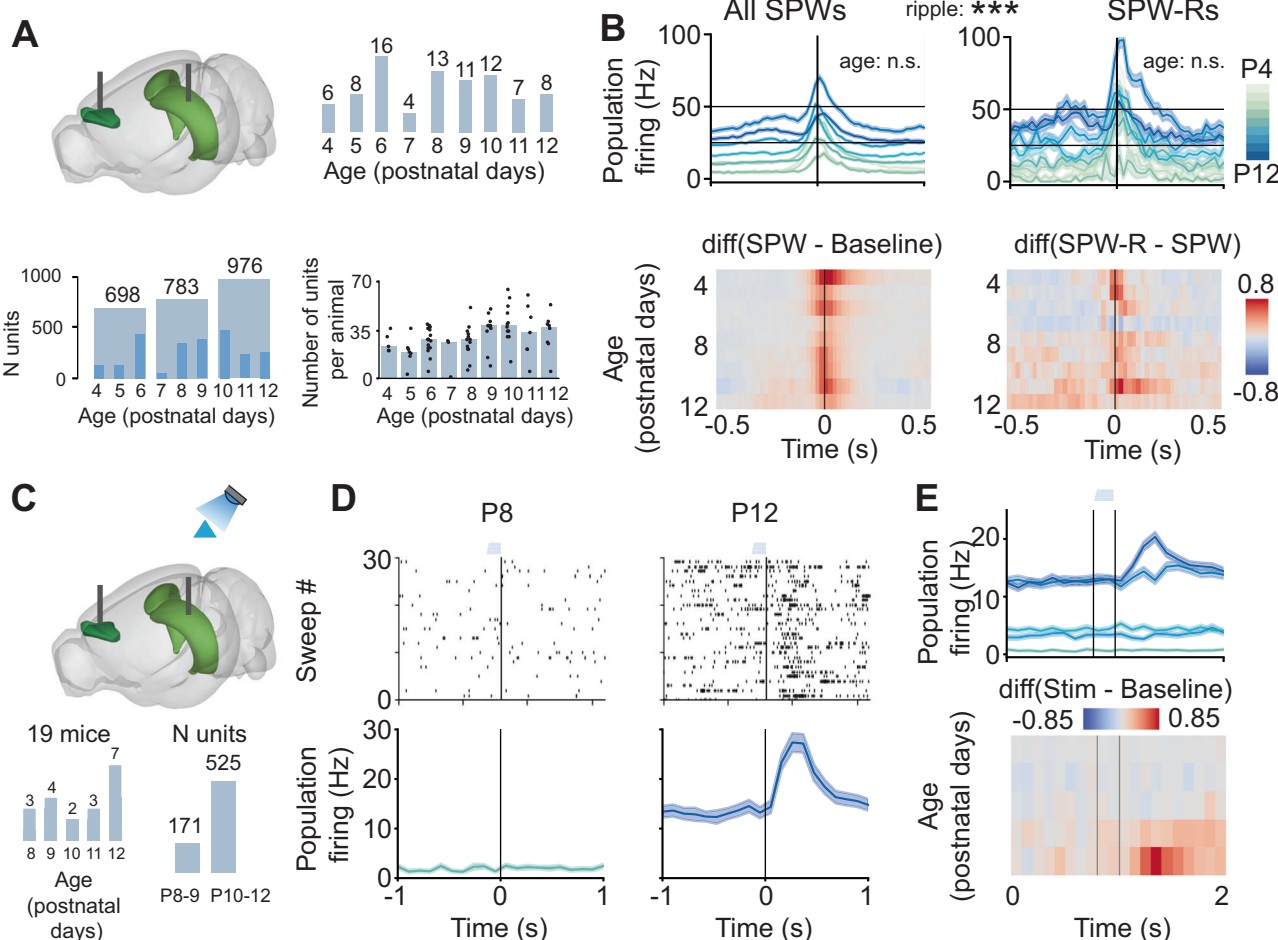

**Fig. 8 | Coupling between the medial prefrontal cortex (mPFC) and HP during spontaneous hippocampal SPWs and optogenetically induced HFO.**
**A** Schematic of the experimental paradigm depicting the location of multi-site electrodes in the hippocampal CA1 area and mPFC area (top left), the total number of investigated mice per age group (top right), distribution of unit numbers over age (bottom left) and mice (bottom right). On the bottom right plot, each dot corresponds to an individual animal, and the bar plot indicates the median number of units. **B** Population firing rates in response to SPW (left) or SPW-R (right) events in the HP CA1, shown as mean ± s.e.m (line plots, top) and the difference between average (mean) activity during SPW and random baseline period/during SPW-R and SPW (heatmaps, bottom). Linear model, firing rate during SPW age slope 95% CI [−0.005; 0.058], $p = 0.108$, firing rate during SPW-R age 95% CI [−0.14; 0.16],

$p = 0.909$; Linear mixed-effects model, mouse as random effect, condition (presence or absence of a ripple), $p < 0.0001$, two-sided. **C** Same as (A) plus optogenetic activation of ChR2-transfected CA1 pyramidal neurons (top), animal number per age group, and the number of single units (bottom). **D** Example of an activity recorded in the mPFC area of P8 and P12 mice. Top: Representative raster plots displaying single unit firing in response to 30 sweeps of tapered-in pulse stimulation (120 ms). Bottom: Population firing rates in response to optogenetic stimulation shown as mean ± s.e.m. Average overall stimulation periods. **E** Population firing rates in response to optogenetic stimulation shown as mean ± s.e.m. (line plot, top) and the difference between average (mean) activity during stimulation periods and random baseline periods (heatmap, bottom). In (B) asterisks indicate a significant effect of age or ripple. **p < 0.01. Source data are provided as a Source Data file.

strong reduction in ripple rate. Our neural network model confirms these findings by showing that, not only are both PYRs and INs necessary to generate ripples, but also that the magnitude of E-I inhibition is a critical factor in the process. If the strength of inhibition exerted by INs onto PYRs falls below a certain level, the model failed to generate ripples. Of note, low levels of I-I conductance do not disrupt periodic activity, but simply impact the frequency of the oscillation. These results suggest that the absence of ripples during early development is due to the lack of adequate levels of inhibition onto PYRs.

This conclusion is supported by the observed 1/$f$ exponent increase from P4 to P10 that documents an age-dependent strengthening of inhibition. Previous studies reported that at the beginning of the second postnatal week, the strength of the perisomatic inhibition onto PYRs in the CA1 region abruptly increases[32,36]. This is particularly important, given that the major source of perisomatic inhibition is PV+-BCs, the class of INs most robustly associated with the generation of ripples in the adult brain. In addition to the developmental dynamics of perisomatic inhibition, several other factors might contribute to the

emergence of ripples at P10. For instance, the duration and propagation time of hippocampal BCs' action potentials decrease throughout development[65]. Similar trends are also displayed in synaptic processes, including faster GABA release and a shorter decay time constant of the corresponding post-synaptic inhibitory current during the first two postnatal weeks[65]. Further, while the model used in the present study consists of only two populations of neurons, the hippocampal CA1 area is populated with more than 21 types of interneurons[19] with different developmental trajectories that might contribute to the emergence of ripples. For example, disinhibition plays an important role in a three-population model of ripple generation[12]. Future investigations should elucidate, how all these factors might affect the developmental changes of ripples.

While the present study monitored the SPW-Rs during the first two postnatal weeks, important developmental milestones shape the hippocampal function later on. For example, hippocampal pre-configured and experience-dependent spiking sequences, representing spatial trajectories in time-compressed way during sleep or rest,

emerge and mature from third-fourth postnatal week on[36,66]. Accordingly, we hypothesize that ripples might also further change during late development. This process might be boosted by increased SPW-R-associated inhibitory inputs to pyramidal cells[67], leading to increased ripple power and rhythmicity, analogously to the developmental changes reported for gamma oscillations in the developing prefrontal cortex[68].

We have previously shown that hippocampal activity is an important early driver of the prefrontal cortex[30,69]. In particular, SPWs elicit a strong response in the oscillatory and firing activity of the developing mPFC, that responds with an increase of spiking activity that begins several hundreds of milliseconds before the SPW, peaks a few milliseconds after, and remains elevated for up to several seconds[30,46]. Here, we confirm and expand these previous results by showing that, already in the second postnatal week, SPW-Rs have a larger impact on prefrontal activity when compared to SPWs. This is of particular interest when considering the role that communication between these two brain areas during SPW-Rs plays in integration of information and long-term storage of memories[70]. While beyond the scope of this manuscript, an intriguing hypothesis that warrants further investigation is the potential role played by the diminished HP-PL communication during SPW (as opposed to SPW-Rs) in the difficulty of storing long-term memories in early infancy[71]. The functional relevance of this question is further stressed by the cognitive deficits that are linked to impaired hippocampal-prefrontal communication in mouse models of mental disorders[47,72–74] and patients affected by these diseases[75,76].

## Methods

### Animals
All experiments were approved by the University Medical Center Hamburg-Eppendorf guidelines and institutional animal welfare officer. All procedures were performed in compliance with German Animal Welfare Act and were approved by the State Authority of Hamburg (Behörde für Justiz und Verbraucherschutz, Amt für Verbraucherschutz, Lebensmittelsicherheit und Veterinärwesen), Germany (N18/015, N19/121). Timed-pregnant mice were housed either individually or in groups of two at a 12 h light/12 hr dark cycle with ad libitum access to water and food at a room temperature of 21 °C and humidity at 43%. The day of vaginal plug detection was defined as embryonic day (E) 0.5, while the day of birth was considered postnatal day (P) 0. Experiments were performed on C57Bl/6J and Dlx5/6-Cre (Tg(dlx5a-cre)1Mekk/J, Jackson Laboratory) mice of both sexes, at the age of P4–12. Each animal underwent a single acute recording session. In accordance with the three Rs guidelines for the use of animals in research, part of the data has been previously acquired in the lab and newly analyzed for the purposes of the present study.

### In utero electroporation
Pregnant mice received additional wet food daily, supplemented with 2–4 drops Metacam (0.5 mg/ml, Boehringer-Ingelheim, Germany) one day before until two days after IUE. At E15.5, pregnant mice were injected subcutaneously with buprenorphine (0.05 mg/kg body weight) 30 min before surgery. Surgery was performed under isoflurane anesthesia (induction 5%, maintenance 2.5%) on a heating blanket. Eyes were covered with eye ointment (Vidisic, Bausch + Lomb, Berlin, Germany) and pain reflexes and breathing were monitored to assess anesthesia depth. Uterine horns were exposed and moistened with warm sterile phosphate buffered saline (PBS). Solution containing 1.25 μg/μl DNA (pAAV-CAG-ChR2(E123T/T159C)2A-tDimer or pAAV-CAG-tDimer) and 0.1% fast green dye at a volume of 0.75–1.25 μl was injected into the right lateral ventricle of embryos using glass capillaries with a sharp and long tip. To target i/v HP we used a tri-polar approach[30,77]. Each embryo was placed between the electroporation tweezer-type paddles (5 mm diameter, both positive poles, Protech,

TX, USA) that were oriented at 90° leftward angle from the midline and a 0° angle downward from anterior to posterior. A third custom-built negative pole was positioned on top of the head roughly between the eyes. Electrode pulses (30 V, 50 ms) were applied six times at intervals of 950 ms controlled by an electroporator (CU21EX, BEX, Japan). Uterine horns were placed back into the abdominal cavity. Abdominal cavity was filled with warm sterile PBS and abdominal muscles and skin were sutured with absorbable and non-absorbable suture thread, respectively. After recovery from anesthesia, mice were returned to their home cage, placed half on a heating blanket for two days after surgery. Fluorophore expression was assessed at P2 in the pups with a portable fluorescence flashlight (Electron microscopy sciences, Hatfield, USA) through the intact skin and skull and confirmed in brain slices postmortem before the animals were used for further analysis.

### Cell quantification
High-resolution single images were captured using a binocular microscope with a resolution of 5760 × 3600 pixels. Image acquisition was performed using a 405 nm laser for DAPI staining and a 568 nm laser for mCherry labeling. Throughout the experiment, consistent image acquisition parameters were maintained across all slices. For each mouse, a single slice containing an electrode trace in the CA1 area was selected. Cellular detection was performed using Cellpose, a state-of-the-art deep learning-based segmentation algorithm implemented in Python 3.8. The detection parameters remained constant for all images, and the accuracy of the results was verified through visual inspection.

### Virus injection
Virus encoding for hM4D(Gi) (AAV9-EF1a-DIO-hM4D(Gi)-mCherry, titer ≥10^14 vg/mL, Plasmid #50461, Addgene, MA, USA) was injected at P0–P2 into HP CA1 under isoflurane anesthesia (0.7 mm anterior to lambda, 2.2 mm lateral to the midline, 0.8 mm deep). During the injection, mice were fixed in a stereotaxic apparatus, and a total volume of 100 nl was administered at a rate of 50 nl/min using a micropump (Micro4, WPI, Sarasota, FL). To prevent any fluid reflux, the syringe was left in place for at least 30 s after injection. Mice were kept on a heating blanket until they fully recovered before being returned to the dam.

### Multisite extracellular recordings in vivo
Recordings were performed unilaterally (right side) in the hippocampal CA1 area and PFC of non-anesthetized P4–P12 mice. The animals were removed from their home cages just before the surgical procedure. Under isoflurane anesthesia (5% induction; 2.5% maintenance) the bone above the right PFC (0.5 mm anterior to bregma, 0.1–0.2 mm lateral to the midline), right HP (1 mm anterior to lambda, 3.5–4 mm lateral to midline depending on age for intermediate/ventral HP CA1; 2.5 mm posterior to bregma, 2.1 lateral to midline for dorsal HP CA1) and cerebellum was removed by drilling holes.

Mice were head-fixed into a stereotaxic apparatus with two plastic bars mounted with dental cement on the nasal and occipital bones. Electrodes (NeuroNexus) were inserted into the PFC (4-shank with 16 recording sites and 100 μm inter-site spacing; depth 1.6–2.0 mm; the angle from the vertical plane 0°) and the HP CA1 (1-shank electrode or optoelectrode with 16 recording sites and 50 μm inter-site spacing; depth 1.3–1.6 mm; the angle from the vertical plane 15–22°). The HP electrode was first inserted 1.3 mm deep and then lowered until the local field potentials (LFP) signal reversal was observed in one of the middle channels. Before insertion, the electrodes were covered with DiI to enable the reconstruction of electrode position post-mortem in brain slices. A silver wire was implanted between skull and brain tissue above the cerebellum and served as ground and reference. Mice were allowed to recover for 20 min. During surgery and recording, mice were kept on a heated blanket at 37 °C.

Extracellular signals were band-pass filtered (0.1–9.0 kHz) and digitized (32 kHz) using a multichannel amplifier and the Cheetah acquisition software (Neuralynx). The length of individual baseline recordings varied between 30 and 117 min (median = 63 min). For each individual postnatal day, the minimum amount of recorded baseline activity aggregated across animals was 9.6 h.

## Optogenetic stimulation

Light stimulation was performed using an Arduino uno (Arduino, Italy) controlled laser system (473 nm wavelength, Omicron, Austria) coupled to a 105 μm diameter light fiber (Thorlabs, NJ, USA) glued to the multisite electrodes, ending 200 μm above the top recording site. At the beginning of each recording session, the response to light stimulation was tested using 3 ms pulses at 16 Hz frequency, and only reliably responding animals were recorded. To induce a high-frequency oscillation (HFO), tapered-in 100 ms square pulses with an initial 20 ms ramp were used[5,17]. Each optogenetic experiment consisted of several trials of increasing light power (minimum number of trials 3, maximum number of trials 13), and every trial contained 30 stimulation periods. The interval between trials was 1 minute. In the first trial, the light power was set to the value determined in light power estimation, and then the light power was gradually increased until either a clear HFO was evoked or one of the two termination criteria was reached: absence of spiking activity or prominent light artifacts in all channels. In the experiments where HFOs were not induced, light power reached at least 50% of max laser power.

## Histology

Mice were anesthetized with 10% ketamine/2% xylazine in 0.9% NaCl (10 mg/g body weight, intraperitoneally) and transcardially perfused with 4% paraformaldehyde. Brains were removed, postfixed in 4% paraformaldehyde for 24 h, and coronally sectioned with a vibratome at 100 μm. Slices were mounted with Fluoromount + DAPI mounting medium. The opsin expression in hippocampal CA1 area and the positions of the DiI-coated electrodes in both CA1 area and PFC were assessed using epifluorescence microscopy.

## Data analysis

Data were analyzed using custom-written Matlab, Python, and R scripts, lab Matlab toolbox (github.com/OpatzLab/HanganuOpatzToolbox), Python Neo toolbox[78], FOOOF toolbox[49], ByCycle toolbox[57], Klusta[79] and Phy (https://github.com/cortex-lab/phy). For the electrode inserted in the CA1 area, seven channels (i.e., the channel with LFP reversal as well as three channels below and three channels above) were used for analysis. For the electrode inserted in the PFC, all 16 channels confined to the PL were used. Recording sites outside PL were discarded.

**Detection of active periods.** Active periods are defined as recorded time windows with oscillatory bursts in the LFP signal. Their detection included signal pre-processing and signal thresholding steps. In the pre-processing step, the extracellular signal from the hippocampal reversal channel was band-pass filtered (4–12 Hz) and downsampled to 250 Hz. Subsequently, the signal was squared, convolved with a boxcar filter (window length 500 ms), and z-scored. The thresholding step included the following substeps: (i) parts of the signal >1 standard deviation or 50 μV were labeled as putative active periods, (ii) periods with inter-period intervals <1 s were merged together, (iii) periods with a peak <2 standard deviations or 100 μV were discarded, and (iv) periods <300 ms were removed.

**Power spectral density.** LFP signals (sampling frequency 32 kHz) from seven channels were band-pass filtered in the 1–500 Hz band (third-order Butterworth filter in a phase preserving manner), downsampled to 1000 Hz, and re-referenced to the average (mean). A battery of narrow notch filters was applied to attenuate the LFP signal at the line frequency (50 Hz) and harmonics. Power spectral density (PSD) was calculated using Welch's method with overlapping windows (window size 1 second, window overlap 250 ms).

**Parameterizing power spectra into periodic and aperiodic components.** The power spectrum in either 70–200 Hz or 25–45 Hz (for high-frequency peak detection and for E-I ratio estimation, respectively) frequency range was parameterized using FOOOF method[49] as a combination of an aperiodic component and putative periodic oscillatory peaks. The aperiodic component was characterized by offset and $1/f$ exponent, whereas peak center frequency, peak bandwidth, and peak power were determined for periodic oscillatory peaks. LFP signals (sampling frequency 32 kHz) from seven channels were band-pass filtered in 1–500 Hz band (third-order Butterworth filter in a phase preserving manner), downsampled to 1000 Hz, and re-referenced to the average (mean). A battery of narrow notch filters was applied to remove the line noise artifacts. PSD was calculated using Welch's method with overlapping windows (window size 1 s, window overlap 250 ms). FOOOF fit was applied on the recording channel located in str. pyramidale (i.e., reversal channel). For high-frequency peak detection, the fit was done using the following parameters: "fixed" aperiodic mode, peak width limits[2,60], and a maximum number of peaks 1. For E-I ratio estimation, the PSD was fit in "fixed" aperiodic mode and with a maximum number of peaks of 0. The fit quality was accessed based on R-squared and only fits with R-squared ≥ 0.95 were included.

**Sharp waves detection.** LFP signals (sampling frequency 32 kHz) from 2 channels were band-pass filtered in the 1–100 Hz band (third-order Butterworth filter in a phase preserving manner) and downsampled to 250 Hz. Sharp waves were detected in the signal acquired by subtracting the signal recorded 100 μm below from the signal recorded 100 μm above the channel in CA1 str. pyramidale. All events that exceeded 3 to 5 standard deviations from the signal mean were labeled as SPWs. The threshold from the interval[3,5] was set for each animal individually and was inversely proportional to the signal standard deviation. A variable threshold was used to compensate for the overall LFP amplitude increase over age. For automatic detection, we used findpeaks Matlab function with the following parameters MinPeakHeight = threshold, MaxPeakWidth = 100 ms, MinPeakProminence = threshold/2, WidthReference = halfheight.

**Ripples detection.** Ripples were detected in 80–200 Hz frequency range using recordings from str. pyramidale (i.e., reversal channel). Two different approaches were used for ripple detection. In the first method, the LFP signal was 1–500 Hz band-pass filtered, downsampled to 1000 Hz, and re-referenced to the average (mean). Subsequently, the LFP signal from the reversal channel was squared and passed through a square filter. Putative ripple periods were detected in 4 consecutive steps. In the first step, signal periods exceeding 3 standard deviations from the signal mean were labeled. Secondly, the labeled periods with in-between intervals shorter than 30 ms were pulled together. In the next step, periods with peaks lower than 6 standard deviations from the signal mean were excluded. Finally, detected oscillatory periods >20 ms and <100 ms (spontaneous ripples) or 120 ms (induced high frequency oscillations) were labeled as putative ripples. In the second method, ripples were detected in the time domain using ByCycle approach[57]. In the first step, the LFP signal from the reversal channel was 1–200 Hz band-pass filtered, downsampled to 2000 Hz, and then segmented into individual cycles. For every cycle, the following features were calculated: amplitude, period, rise-decay symmetry, and peak-trough symmetry. In the second step, for every cycle, it was determined whether the cycle was part of an oscillatory period. A cycle was labeled as a part of an oscillatory activity if its amplitude and period were relatively consistent with the adjacent

cycles (amplitude consistency threshold and period consistency threshold), and if its rise and decay flanks were rather monotonic (monotonicity threshold). The thresholding parameters for oscillatory periods detection were adjusted on an animal-by-animal basis, and the analysis was run with a range of thresholding parameters to ensure that results were not dependent on one particular set of parameters. Only ripples detected by both approaches and co-occurring with SPWs were considered for further analysis. Co-occurrence with SPWs was defined as the non-zero overlap between ripple time interval and SPW time interval where SPW time interval is [SWP maximum amplitude in radial layer +- halfwidth/2]. Different frequency bands: 100–200 Hz, 100-300 Hz, 80-250 Hz (as for[17]) and 100-250 Hz (as for[56]) have been used to confirm the ripple detection.

**Single-unit activity.** Spike sorting was performed using Klusta and automatically detected clusters were then manually curated using Phy.

**Movement periods detection.** During in vivo electrophysiological experiments, mice were recorded with a monochrome camera (UI-3360CP-NIR-GL R2, iDS imaging, Germany) under low light conditions. Videos were acquired with the uEye Cockpit software (iDS imaging) with a resolution of 2048 × 1088 pixels and a frame rate of 14.3 frames per second. Synchronization between the video stream and electrophysiological recording was performed with light pulses. The mouse position was tracked with DeepLabCut, a deep neural network-based methodology[80]. A neural network was trained over 200,000 iterations to accurately monitor six reference points on the mouse: the tail base, tail tip, hind left leg, hind right leg, left head fixation bar, and right head fixation bar. We denoted as "movement" periods those in which the tracked reference points exhibited a change in position of more than 25 pixels across consecutive frames.

**Statistical analysis.** Statistical analysis was done in R. Non-nested data were analyzed with (generalized) linear models and nested data with (generalized) linear mixed-effects models with the mouse as a random effect. Non-monotonic data were fit with piece-wise linear regression models using segmented R package. Initial model selection was based on exploratory analysis of the data, and then the goodness of fit was evaluated using explained variance (R-squared) and residuals distribution (check_model function from performance R package). When several models were fit on the same data, the model fit was compared using compare_performance function from performance R package. 95% confidence intervals were computed using confint R function, p-values for linear mixed-effects models were computed with the lmerTest R package, post hoc analysis with Tukey multiple comparison corrections was done using emmeans R package, and the model fit was plotted using plot_model function from sjPlot R package.

**Neural network model**
**Model description.** The spiking neural network was implemented similarly to[9]. The detailed description is given in Supplementary Table S1. The network had no spatial structure, and neurons were randomly connected with connection probability $p_{ij}$ where j was a source and i was a target. Self-connections, as well as multiple connections between any pair of neurons, were not allowed. For each pair of neurons from the source and the target populations, a connection was created if a random number was smaller than the connection probability for the pair.

**Model parameters.** The "default" model was constrained with biologically plausible parameters for adult CA1 circuitry. The number of neurons in E and I populations and connectivity in the network were estimated based on a quantitative assessment of CA1 local circuits[19]. The model was set to represent a network in a 0.4 mm thick hippocampal slice, accounting for 4% of the total hippocampal volume. The total number of PV⁺ BCs in the hippocampus is estimated to be 5530, yielding 221 cells in the slice. A slightly lower estimate is obtained based on the PV⁺ cell density in the CA1 pyramidal layer. The density is calculated to be $5.4 \times 10^3$ cells/mm$^3$ [54], and given that 60% of PV⁺ cells are BCs, the number of PV⁺ BCs in slice is 184. The number of I cells was set to 200, the average between the two estimates. Given that the ratio between PYRs and PV⁺ BCs is 60:1, the number of PYRs was set to 12000. The connection probabilities were set to $p_{EE} = 0.0164$, $p_{IE} = 0.1$, $p_{EI} = 0.1$, $p_{II} = 0.2$, in the agreement with published ripples models[9,11].

The parameters for individual neurons (Supplementary Table S2) and synapses (Supplementary Table S3) were set to be in an agreement with experimental literature and the two before-mentioned published ripple models.

**Developmental change in inhibition.** In order to assess the effect of strengthening inhibition on the network activity, we modulated inhibitory synaptic conductances through multiplication with values from 0.095 to 2 for E-I connections (c_gei × $g_{inh,peak}^E$) and from 0.05 to 2 for II connections (c_gii × $g_{inh,peak}^I$). The start conductance multiplier was the lowest that prevented the runaway activity in the network. Multiplying with 1 corresponds to an adult-like network. The summary of tested changes is given in Supplementary Table S4.

**Simulations.** Simulations were performed using Brian 2 simulator for spiking neural networks[81] and parameter exploration toolkit pypet (https://github.com/SmokinCaterpillar/pypet). For every explored parameter combination, 30 simulations with the independent random realization of connectivity and membrane noise were performed to test whether the oscillations, generated by the network, were robust. For every run, all except one parameter were fixed on adult values. The standard deviation of the membrane noise σ was set to 5 mV. At this noise level spiking activity in the network was present in the absence of external input. The network was simulated for a duration of 500 ms in test runs with adult parameters and for a duration of 100 ms in runs with modeled developmental changes. At the beginning of a simulation, neurons were initialized with voltages from Gaussian distribution with a mean equal to the resting potential and standard deviation equal to 0.1 mV. All simulations were performed with a time step (dt) of 0.1 ms, and as an integration method, Euler-Maruyama algorithm was used. Population firing rates were computed as averaged across neurons' instantaneous firing rates, with a Gaussian smoothing window with a width of 0.5 ms. PSD was calculated using a median filter of the squared FFT (Fast Fourier Transform) magnitude.

**Reporting summary**
Further information on research design is available in the Nature Portfolio Reporting Summary linked to this article.

## Data availability
The LFP and SUA data generated in this study have been deposited in the G-Node GIN under accession code https://gin.g-node.org/iinnpp/ripples_emergence_inhibition (https://doi.org/10.12751/g-node.bcds1m). Source data are provided with this paper.

## Code availability
The main analysis code to generate the results is described in the method section and/or is available at following open-access repositories: https://github.com/OpatzLab/HanganuOpatzToolbox/ and https://github.com/iinnpp/spwr_snn (https://doi.org/10.5281/zenodo.10424461).

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

## Acknowledgements

We thank A. Dahlmann and P. Putthoff for excellent technical assistance as well as Drs. Lingzhen Song, Xiaxia Xu, Sebastian Bitzenhofer, Johanna Kostka, Jastyn Pöpplau, and Jan Marker for valuable discussions. This work was funded by grants from the German Research Foundation (Ha4466/11-1, Ha4466/20-1 and SFB 936 B5 to I.L.H.-O.), European Research Council (ERC-2015-CoG 681577 to I.L.H.-O.), Horizon 2020 MSCA-ITN (860563 to I.L. H.-O.), DEEPER (101016787 to I.L.H.-O.), Landesforschungsförderung Hamburg (LFF73 and LFF76 to I.L. H.-O.), the LOEWE CePTER – Center for Personalized Translational Epilepsy Research (to T.M.S.) and Johanna Quandt Foundation (to J.T.).

## Author contributions

Conceptualization: M.C., I.L. H.-O. Methodology: I.P., M.C., I.L. H.-O., T.M.S., J.T. Investigation: I.P., M.C. Visualization: I.P. Supervision: M.C., I.L. H.-O., T.M.S., J.T. Writing—original draft: I.P. Writing—review & editing: I.P., M.C., I.L. H.-O., T.M.S., J.T.

## Funding

## Competing interests

The authors declare that they have no competing interests.
