## [Peer Review File · Nature Communications]

REVIEWER COMMENTS

Reviewer #1 (Remarks to the Author):

Pochinok and colleagues chart the developmental emergence and circuits dynamics of sharp-wave ripples in the hippocampal CA1 area. This study is highly timely. A concerted research effort over the past decade has been dedicated to elucidating the mechanism and function of SWRs in adult animals, yet research on the development of this unique synchronicity event has been scant. To address this caveat, Pochinok et al. provide a highly comprehensive analysis of the development of SWRs during the first and second post-natal week. They show that the fast oscillatory component of SWRs only emerges during the end of the second post-natal week and based on simulations and physiological proxies for E-I balance show their emergence is likely dependent on the maturation of peri-somatic inhibition. Finally, they show SWRs are associated with heightened HPC-PFC interactions.

I am enthusiastic about this study and I believe the results would be of broad interest to the hippocampal and developmental neuroscience community. I have a number of comments however about their methods and controls which need to be addressed. These should be straight forward to do with their current data and would substantially increase the strength of their results. I detail my comments below.

- The frequency band used to detect ripples seems rather low (i.e. 80-200Hz). 80Hz is really in the high gamma range. A more standard frequency band starts >100Hz and goes up to 200-300 Hz. As the ripple frequency is not thought to change in development (as the authors show, as well as Buhl & Buzsaki (2005)), I see no reason to use such a low filter band. Further, starting the filter band in the high gamma range makes the interpretation of the authors' results somewhat unclear, as some of their ripple events could be high gamma periods. Could the authors repeat their main analysis with a higher frequency filter?
- Related to SWR detection, I don't see any movement controls. Normally, when detecting ripples, researchers exclude events when the animal is moving. I appreciate these recordings are carried out while the animals are head-fixed. Nonetheless, animals can move while head fixed (e.g. they can move their legs, they may struggle on the rig, they may groom) and/or they may be highly aroused. The authors need to do something to eliminate events that occur during movement/high arousal, as these are likely not genuine SWR events. If the researchers do not have access to movement related data they could try to try limiting their analysis to NREM sleep periods.
- I appreciate how thoroughly the authors have analysed SWRs in their study. However, in addition to the analyses of frequency, duration, number of spikes, etc. that the researchers have done, it would be interesting to also analyse how the number of cells participating in SWRs changes with age, the number of ripple events without a SPW (e.g. Navas-Olive et al. (2022)) and the proportion of long-duration/multi-bout SWRs (Oliva et al. (2018)). This last point may be particularly interesting as long-duration SWRs

(that are likely composed of multiple bouts of ripples) are thought to be particularly relevant for memory (Fernandez-Ruiz(2019)). Given how large the authors' dataset it presents a unique opportunity to characterise the development of SWRs in high detail.

- Finally, I would caution the authors to not refer to the SWRs they see in their P10-P12 pups as adult-like. They do not have adult animals to compare to and it is highly unlikely that SWRs by the end of the 2nd post-natal week are mature (e.g. Buhl & Buzsaki (2005)). A more conservative interpretation is that SWRs *start* to emerge at this age.

Reviewer #2 (Remarks to the Author):

The article by Pochinok et al reports extremely interesting and carefully executed experiments focused on the development of sharp-wave ripple complexes during post-natal development in the mouse.

The report is written in a very clear and engaging manner, and its review of the literature and conceptual landscape on this topic is commendable.

What I offer below are suggestions to further increase the interest of the results presented and aid their interpretation.

- The main strength of the article lies in the careful characterisation of the emergence of ripples (figs 3 and 4) – and the results clearly point towards a relatively abrupt increase in detectable ripples from P10 onwards. The data presented are convincing. I have only one suggestion regarding this set of experiments – in several panels of figure 4 (e.g. C, E, F) the datapoints related to P9 seem to fall into a bimodal distribution. Would it be possible/fruitful to analyse at least some of the data presented in the manuscript with animal as a unit of investigation, rather than neuron? This approach might be particularly informative in the context of development where some inter-individual “jitter” across the age domain is to be expected.

- Another consideration relates to the detection of ‘short’ ripples (<30ms duration). Data presented in figure 4F seem to suggest that there is a specific developmental period during which these short ripples are detected (P6-P8) – could the authors comment on the potential origin of these ‘short’ ripples and why they would only be detectable during this time period? (Do authors think that this is potentially interesting, or could this be attributed to a technical issue to do with the detection methods employed here?)

- Could authors comment on whether the percentage of SPWs with ripples detected at P12 (oldest age sampled; fig 4E) is consistent with that observed in adult mice/rodents? If not, what other mechanisms do the authors posit for any further maturation of ripples beyond P12?

- More generally, could authors comment on whether they think their SUA activity captures both PYR and IN firing? If yes, is there any way they could try and analyse the data to separate the contribution of PYR and IN firing?

- Data presented in fig 6 (optogenetic activation of CA1 resulting in ripple like fast frequency oscillations) are robust. However, we noticed that the authors 'slip' in the text by referring to these events as bona fide ripples (results section, line 437; first paragraph of the discussion, line 470). As the activity was evoked using optogenetic stimulation, the oscillations produced are not bona fide ripples. I would be satisfied with 'ripple-like' as a more accurate way of referring to these oscillations.

- The use of DREADDs to inhibit hippocampal IN is a little bit of a "blunt" manipulation - a more time-controlled manipulation would have been more persuasive and informative. Did authors notice seizure activity in the mice? Could the authors comment on why this silencing method was chosen, and its limitations? How long did the silencing effects last for?

- Another question/comment relates to the approach taken to study the effect of ripple emergence in hippocampus on mPFC. Why was this only done using optogenetics, rather than observing the natural occurrence of ripples and studying their effects in mPFC?

- More generally, we assume that each animal only contributed one recording session to the dataset (one age datapoint) – could this be clarified and made more explicit in the methods section?

Copy editing changes:

- From lines 424-436 reference to figure 7 should be replaced by reference to figure 8.

- Line 522 "strengthen" should be changed to its gerundive "strengthening".

- Sentence in line 528 is mangled and should be corrected.

Reviewer #3 (Remarks to the Author):

The authors of this study combined *in vivo* electrophysiology with optogenetics and chemogenetics in developing P4-P12 mice as well as *in silico* models to address the relationship between the developmental increase of inhibition and the emergence of the hippocampal ripples. By *in silico* simulation, they showed theoretically that increased inhibition to CA1 excitatory units promotes the emergence of ripples. By electrophysiological recording, they showed that the age-dependent increase of inhibition ratio in CA1 stabilized at the similar age as the emergence of spontaneous and light-induced ripples (P10), and they confirmed the stronger inhibition to pyramidal neurons in older ages. Using chemogenetics, they showed that silencing interneurons in CA1 decreased SPW-Rs rate in P11-P12 mice. Finally, the authors found that the firing rate increase in the downstream medial prefrontal cortex area was larger during spontaneous SPW-Rs and optogenetically induced CA1 high-frequency oscillations than during SPW events alone. Given all these results, the authors concluded that the development of inhibition promotes the emergence of hippocampal ripples in the third week of postnatal life. This is an interesting developmental study aiming to clarify the age at which the hippocampal CA1 ripple activity and its impact on downstream areas emerge, with implications for the emergence of memory consolidation as well as of several additional cognitive processes. I have several comments and recommendations aimed at improving the manuscript, as listed below.

1. The main conclusion was largely drawn from the co-occurrence of the increased inhibition and emergence of ripples (Both on P10). This co-occurrence is more of a correlation, though the chemogenetic analysis implied some causal relationship. The authors should add some discussion about the correlational versus causal relationship between inhibition and ripple developments.
2. The strength of inhibition was indirectly measured by models or population temporal firing profile, and it might be difficult to generate a direct perception of how the inhibition develops across ages. It might be more powerful if the authors can distinguish PYRs and INs from single unit level by the firing properties in individual animals and directly show the developmental profiles separately.
3. It seems that there is a broad distribution of SPW-Rs rate and the ratio of SPWs with ripples across mice on P11 and P12 (from figure 4D, E). I would be stronger if the authors could show the individual mice differences in the ripples and inhibitions, and whether there is a correlation between ripples and inhibitions across individuals, which would support their conclusion.
4. The authors discuss early development of the hippocampus (weeks 1-2) and then jump to discussing adult data and implications. There is a small but highly relevant literature on the development of ripple activity on postnatal weeks 3-4 that would be quite important to the current study, and which should be discussed. In particular, the study of Noguchi, Matsumoto, Ikegaya, *JNeurosci* 2023 that already showed the relationship between the maturation of inhibition and of ripple activity in CA1 *in vivo* in mice, and

the study of Farooq and Dragoi, Science 2019 that showed further maturation of ripple activity in freely moving rats, both studies of neural activity during developmental postnatal weeks 3-4.

5. Can the authors investigate other properties of ripples, such as ripple doublets and temporal timing details of the hippocampal-prefrontal interactions?

6. The computational model used in Figure 2 is used as justification for the remaining experiments. While that might have been indeed the intuition for this study, I would recommend moving the model toward the end of the manuscript (last figure) and presenting all the experimental results first.

7. At some earlier ages during neurodevelopment, the features of ripples (lower frequency, fewer cycles) resemble those of other oscillation, such as the high-gamma oscillation. The authors should discuss their reasons for excluding this possible confound between ripple and high-gamma oscillations in their data.

8. At least one study, Stark, Roux, Eichler, Buzsaki, PNAS 2015 reported induction of ripples in the adult CA1 by driving putative pyramidal neurons alone. The authors should discuss how those results compare to their conclusion that direct manipulation of inhibitory activity is necessary for artificial ripple creation.

Minor comments

1. The condition of mice while recording should be addressed, e.g., recording while they were resting in cages or somewhere else.

2. Line 528 '..... throughout development (Doischer et al., 2008)' the citation format is different from other citations.

3. There seems to be some missing left parenthesis or redundant right parenthesis in Line 212-215, which made it difficult to understand the sentence.

4. It would be more reliable if statistical measurements can be added on figure 4G and 4H, Figure 5D, and Figure 6E.

Reviewer Comments:

Reviewer #1:

Pochinok and colleagues chart the developmental emergence and circuits dynamics of sharp-wave ripples in the hippocampal CA1 area. This study is highly timely. A concerted research effort over the past decade has been dedicated to elucidating the mechanism and function of SWRs in adult animals, yet research on the development of this unique synchronicity event has been scant. To address this caveat, Pochinok et al. provide a highly comprehensive analysis of the development of SWRs during the first and second post-natal week. They show that the fast oscillatory component of SWRs only emerges during the end of the second post-natal week and based on simulations and physiological proxies for E-I balance show their emergence is likely dependent on the maturation of peri-somatic inhibition. Finally, they show SWRs are associated with heightened HPC-PFC interactions.

I am enthusiastic about this study and I believe the results would be of broad interest to the hippocampal and developmental neuroscience community. I have a number of comments however about their methods and controls which need to be addressed. These should be straight forward to do with their current data and would substantially increase the strength of their results. I detail my comments below.

We thank the reviewer for the constructive feedback and helpful comments and suggestions. We addressed them point by point below and modified the manuscript text and figures accordingly.

1. The frequency band used to detect ripples seems rather low (i.e. 80-200Hz). 80Hz is really in the high gamma range. A more standard frequency band starts >100Hz and goes up to 200-300 Hz. As the ripple frequency is not thought to change in development (as the authors show, as well as Buhl & Buzsaki (2005)), I see no reason to use such a low filter band. Further, starting the filter band in the high gamma range makes the interpretation of the authors' results somewhat unclear, as some of their ripple events could be high gamma periods. Could the authors repeat their main analysis with a higher frequency filter?

We followed the reviewer's suggestion and repeated the analysis shown in Fig. 4 (i.e., the SPW-R detection in the temporal domain) using higher frequency filters. We added four additional frequency bands: 100–200 Hz, 100-300 Hz, 80-250 Hz (used for ripple detection by Stark et al. 2014) and 100-250 Hz (recommended by Liu et al. 2022 as a ripple frequency band criterion for rodents). Comparing the SPW-R rate obtained using the original frequency filter (Fig.4D) with SPW-R rates calculated with the four additional frequency filters (fig. S5A), we detected a similar effect of age on SPW-R rate, namely, no change from P4 to P10, followed by an increase from P10 to P12 (fig. S5B). The detailed statistical results are summarized in the table below:

frequency filter	break-point location	P4-P10 slope	P10-P12 slope
80-200Hz	9.75, 95% C.I. [8.69; 10.82]	age slope = -0.0028, 95% C.I. [-0.17; 0.16], p = 0.974	age slope = 0.80, 95% C.I. [0.36; 1.23], p = 0.00096
100-200Hz	9.73, 95% C.I. [8.62; 10.84]	age slope = 0.0147, 95% C.I. [-0.06; 0.09], p = 0.691	age slope = 0.36, 95% C.I. [0.16; 0.57], p = 0.00165
100-300Hz	9.66, 95% C.I. [8.59; 10.73]	age slope = 0.0012,	age slope = 0.317,

		95% C.I. [-0.06; 0.06], p = 0.970	95% C.I. [0.14; 0.49], p = 0.00098
80-250Hz	9.85, 95% C.I. [8.94; 10.76]	age slope = 0.0143, 95% C.I. [-0.05; 0.08], p = 0.664	age slope = 0.38, 95% C.I. [0.20; 0.56], p = 0.00022
100-250Hz	9.70, 95% C.I. [8.63; 10.76]	age slope = 0.0072, 95% C.I. [-0.06; 0.07], p = 0.820	age slope = 0.322, 95% C.I. [0.15; 0.49], p = 0.00096

In addition, we evaluated the effect of age and frequency band on the SPW-R rate and found that, while age alone has a significant effect on SPW-R rate, the frequency filter alone does not affect SPW-R rate in a significant way (age effect, P11 = 0.926, 95% CI [0.63; 1.23], $p < 10^{-7}$; age effect, P12 = 0.924, 95% CI [0.64; 2.21], $p < 10^{-8}$; frequency band effect, 100-200Hz = -0.023, 95% CI [-0.05; 0.008], $p = 0.147$; frequency band effect, 100-300Hz = 0.007, 95% CI [-0.02; 0.04], $p = 0.655$; frequency band effect, 80-250Hz = -0.029, 95% CI [-0.06; 0.001], $p = 0.067$; frequency band effect, 100-250Hz = -0.020, 95% CI [-0.05; 0.01], $p = 0.193$; linear mixed-effect model with interactions) (fig. S5C). Lastly, we compared the results of SPW-R detection in the temporal domain (Fig. 4D, fig. S5A) with the detection results in the frequency domain (Fig.3F) and showed that the presence of oscillatory peak in power spectrum significantly correlates with the SPW-R rate for all frequency filters (peak effect, 80-200Hz = 0.72, 95% CI [0.38; 1.06], $p < 10^{-4}$; peak effect, 100-200Hz = 0.58, 95% CI [-0.27; 0.88], $p = 0.00029$; peak effect, 100-300Hz = 0.47, 95% CI [0.21; 0.73], $p = 0.00056$; peak effect, 80-250Hz = 0.58, 95% CI [0.31; 0.86], $p = < 10^{-4}$; peak effect, 100-250Hz = 0.48, 95% CI [0.22; 0.74], $p = 0.000402$; linear model) (fig. S5D).

We added the results to the manuscript (lines 217-221, 763-764) and included a new supplementary figure (Fig. S5). The statistics for the new results were added to the Supplementary Table S5.

2. Related to SWR detection, I don't see any movement controls. Normally, when detecting ripples, researchers exclude events when the animal is moving. I appreciate these recordings are carried out while the animals are head-fixed. Nonetheless, animals can move while head fixed (e.g. they can move their legs, they may struggle on the rig, they may groom) and/or they may be highly aroused. The authors need to do something to eliminate events that occur during movement/high arousal, as these are likely not genuine SWR events. If the researchers do not have access to movement related data they could try to try limiting their analysis to NREM sleep periods.

We agree with the reviewer that, in the used recording configuration, the head fixation with two bars mounted on the nasal and occipital bones prevents head movement and limits movement of the frontal body part. However, mice can still freely move their hind legs. To exclude the confounding effects of movements on the event detection, we recorded the activity from the HP CA1 area of non-anesthetized P11-P12 mice (n=7, 3 P11 mice, 4 P12 mice) simultaneously with video tracking of mouse position using DeepLabCut (Mathis et al. 2018). A neural network was trained over 200,000 iterations to accurately monitor six reference points on the mouse: the tail base, tail tip, hind left leg, hind right leg, left head fixation bar, and right head fixation bar. We denoted as "movement" periods those in which the tracked reference points exhibited a change in position of more than 25 pixels across consecutive frames. The movement periods were aligned with the detected SPW-R events. We detected no difference in SPW-R rate between full recordings and recordings with removed movement periods (condition effect = -0.13, 95% CI [-1.49; 1.23], $p = 0.837$) (fig. S6A). On the individual mouse level, the average decrease in SPW-R rate was minimal: 0.08 SPW-R/min on an average SPW-R rate of 1.8 SPW-R/min. Overall, mice rarely moved, as the average moving time

was only 5% of the total recording duration (fig. S6B). We added the new results to the manuscript (lines 222-225, 769-779) and included a new supplementary figure (Fig. S6). The statistics for the new results were added to the Supplementary Table S5.

3. I appreciate how thoroughly the authors have analysed SWRs in their study. However, in addition to the analyses of frequency, duration, number of spikes, etc. that the researchers have done, it would be interesting to also analyse how the number of cells participating in SWRs changes with age, the number of ripple events without a SPW (e.g. Navas-Olive et al. (2022)) and the proportion of long-duration/multi-bout SWRs (Oliva et al. (2018)). This last point may be particularly interesting as long-duration SWRs (that are likely composed of multiple bouts of ripples) are thought to be particularly relevant for memory (Fernandez-Ruiz(2019)). Given how large the authors' dataset it presents a unique opportunity to characterise the development of SWRs in high detail.

We followed the reviewer's suggestion and broadened the analysis by quantifying long-duration/multi-bout SPW-Rs (fig. S7A) and "solo" ripple events (fig. S7B). In addition, we expanded the analysis of spiking activity during SPWs by quantifying the number of concurrently engaged neurons in SPW events (fig. S8A), along with an analysis of the reliability of a unit's involvement, measured as the percentage of SPWs in which a specific unit contributed at least one spike (fig. S8B). For the investigation of age-related changes in ripple duration, we adopted the approach outlined by Oliva et al. 2018. Ripples were categorized into three groups based on their duration: 30-50, 50-80, and 80-100 ms. Notably, the study also defined a fourth group including durations > 100 ms, which we excluded due to the ripple detection approach's upper limit cutoff at 100 ms, resulting in the exclusion of events surpassing this threshold. We did not individually identify multi-bout ripple events; the detection method combines events closer than 30 ms, meaning any multi-bout events present were grouped into the 50-80 or 80-100 ms categories. We observed an increase in SPW-R rates in the 30-50 ms and 50-80 ms groups with age (30-50 ms age slope = 0.060, 95% CI [0.036; 0.085], $p < 10^{-5}$, linear model; 50-80 ms age slope = 0.026, 95% CI [0.016; 0.036], $p < 10^{-6}$, linear model) (fig. S7A, left and middle), yet no significant change in the 80-100 ms group (age slope = 0.0016, 95% CI [-0.001; 0.004], $p = 0.2348$, linear model) (fig. S7A, right). Of note, SPW-R rate in the 80-100 ms group was notably lower compared to the other groups. Since long duration ripples are known to be particularly relevant for memory, the low rate and absence of a developmental trend in the 80-100 ms group for P4-12 mice might reflect the lack of hippocampal sequences that emerge not before the third and fourth postnatal weeks.

Consistent with Navas-Olive et al. 2022, we detected "Ripples no-SW" events across all ages, with their rate showing an age-related increase (age slope = 0.183, 95% CI [0.073; 0.293], $p = 0.00129$, linear model) (fig. S7B).

The new spiking activity analysis revealed a developmental increase in cell engagement during SPW events. Specifically, we observed an age-related rise in the number of units contributing to these events (age slope = 0.414, 95% CI [0.296; 0.531], $p < 10^{-9}$, linear mixed-effect model) (fig. S8A). Notably, while units participating in roughly half of all SPWs were already present at younger ages, the reliability of their participation increased with age (age slope = 1.13, 95% CI [0.603; 1.658], $p < 10^{-4}$, linear mixed-effect model) (fig. S8B).

We added the new results to the revised manuscript (lines 239-242, 295-298) and added two new supplementary figures (Figs. S7 and S8). The statistics for the new results was added to the Supplementary Table S5.

4. Finally, I would caution the authors to not refer to the SWRs they see in their P10-P12 pups as adult-like. They do not have adult animals to compare to and it is highly unlikely that SWRs by the end of the 2nd post-natal week are mature (e.g. Buhl & Buzsaki (2005)). A more conservative interpretation is that SWRs *start* to emerge at this age.

In line with the suggestion, we removed the “adult-like” descriptor (line 243-244).

Reviewer #2:

The article by Pochinok et al reports extremely interesting and carefully executed experiments focused on the development of sharp-wave ripple complexes during post-natal development in the mouse.

The report is written in a very clear and engaging manner, and its review of the literature and conceptual landscape on this topic is commendable.

What I offer below are suggestions to further increase the interest of the results presented and aid their interpretation.

We thank the reviewer for the constructive feedback and helpful comments and suggestions. We addressed them point by point below and modified the manuscript text and figures accordingly.

1. The main strength of the article lies in the careful characterisation of the emergence of ripples (figs 3 and 4) – and the results clearly point towards a relatively abrupt increase in detectable ripples from P10 onwards. The data presented are convincing. I have only one suggestion regarding this set of experiments – in several panels of figure 4 (e.g. C, E, F) the datapoints related to P9 seem to fall into a bimodal distribution. Would it be possible/fruitful to analyse at least some of the data presented in the manuscript with animal as a unit of investigation, rather than neuron? This approach might be particularly informative in the context of development where some inter-individual “jitter” across the age domain is to be expected.

As pointed by the reviewer, we observed inter-individual variability across age. This might be due to (i) the fact that the birth time of mice is given ± 12 hours and (ii) the fact that the recordings were not all performed at the same time during the day. Therefore, it is plausible that some P9 recordings exhibit proximity to P8, while others are rather closer to P10 recordings, causing the observed bimodal distribution. To account for inter-individual variability in the performed analysis and used statistical models, we either considered data on the animal level or included “animal” as random effect in mixed-effects models while working with data on the neuron level or SPW-R event level. The data on Fig.4C, D, E and F are presented on the mouse level, namely, the dots correspond to individual animal values. The statistical models for the data on Fig.4C, D, E are designed on the animal level while the model for data on Fig.4F (ripple length) is designed on the ripple event level with animal as a random effect in the mixed-effect model. Thus, we never “inflated” the number of statistical units beyond the actual number of recorded mice.

2. Another consideration relates to the detection of ‘short’ ripples (<30ms duration). Data presented in figure 4F seem to suggest that there is a specific developmental period during which these short ripples are detected (P6-P8) – could the authors comment on the potential origin of these ‘short’ ripples and why they would only be detectable during this time period? (Do authors think that this is potentially interesting, or could this be attributed to a technical issue to do with the detection methods employed here?)

Short ripples were consistently detected across all investigated age groups. In P4-P9 mice, they represented ~50% of all SPW-R events (Fig. 4H), resulting in an average ripple length of 32 ms within that group. The raw data displayed in Fig. 4G might suggest higher numbers of short SPW-Rs in P6-P8 ages, yet normalizing the values to the number of animals within each age group (Fig. 1A: P4 9 mice, P5 9 mice, P6 16 mice, P7 10 mice, P8 17 mice) led to a constant fraction of short

ripples for the investigated time window. However, it is questionable whether these events represent true oscillations, since short ripples had maximum three oscillatory cycles. In adult HP, events classified as ripples typically exhibit 3-9 oscillatory cycles (Buzsáki 2015).

Formal unbiased criteria to exclude the short ripples identified by the used detection approach (SPW detection combined with two distinct ripple detection methods) are missing. Although the detection method was robust across frequency bands (fig. S5A) and resulted in statistically indistinguishable SPW-R rate developmental trends (fig. S5B), we cannot definitively rule out the presence of detection artifacts. One common cause of false positives in ripple detection is population synchrony (Navas-Olive et al. 2022, Liu et al. 2022). Therefore, it is possible that the short non-rhythmic spike bursts occurring upon the arrival of the sharp waves might have been detected as short ripples due to filtering effects. We briefly discussed this limitation of the approach in the manuscript (lines 232-234).

3. Could authors comment on whether the percentage of SPWs with ripples detected at P12 (oldest age sampled; fig 4E) is consistent with that observed in adult mice/rodents? If not, what other mechanisms do the authors posit for any further maturation of ripples beyond P12?

Several animal-related variables (e.g., arousal, sleepiness, circadian rhythm) and technical factors (e.g., detection methods, frequency band, amplitude thresholds) hamper reliable comparisons between SPW-Rs reported by different studies. The event occurrence appears highly variable across studies, ranging from 0.01 to > 10 Hz (Liu et al. 2022). For instance, one study reported 1.9 events per minute by incorporating a sharp wave in the ripple detection criteria (Jiang et al. 2020), while another reported 10–40 events per minute without requiring a sharp wave for ripple detection (Norman et al. 2019). Since sharp waves are often used as a criterion in detecting ripples, instances of sharp waves without accompanying ripples are not commonly reported in adult data.

Investigation of SPW-Rs in P2-18 rats showed that the likelihood of observing ripples with sharp waves continues to rise until around P14, when it reached adult levels of ~50-60% (Mohns et al. 2007). Given that the hippocampus undergoes important functional maturation until third-fourth postnatal week (Farooq and Dragoi 2019), it is likely that the amplitude and power of ripples as well as the occurrence of SPWs with ripples will increase.

In line with the suggestion of the reviewer, we added a discussion of the SPW-Rs development during the third and fourth postnatal weeks (lines 563-570).

4. More generally, could authors comment on whether they think their SUA activity captures both PYR and IN firing? If yes, is there any way they could try and analyse the data to separate the contribution of PYR and IN firing?

As highlighted by the reviewer, PYRs and INs might be differently involved during SPW-Rs, similarly to previous data from adult rodents (Stark et al. 2014). While most recorded units are putatively PYRs, due to the 60 PYRs:1 INs ratio in the HP CA1 (Bezaire and Soltesz 2013), understanding the contribution of both neuronal populations to the SPW-Rs development would be of critical relevance. The commonly used approach to classify the two types of units relies on difference in a spike waveform shape with fast-spiking (putative PV+) INs having a narrower one. However, during early development, the IN waveform is undistinguishable from the PYR waveform (Weir et al. 2014; Chini et al., 2022). Thus, separating the contribution of PYR and IN firing at this age is not possible.

5. Data presented in fig 6 (optogenetic activation of CA1 resulting in ripple like fast frequency oscillations) are robust. However, we noticed that the authors 'slip' in the text by referring to these events as bona fide ripples (results section, line 437; first paragraph of the discussion, line 470). As the activity was evoked using optogenetic stimulation, the oscillations produced are not bona fide ripples. I would be satisfied with 'ripple-like' as a more accurate way of referring to these oscillations.

As recommended, we rephrased to "ripple-like" (lines 463, 496-497).

6. The use of DREADDs to inhibit hippocampal IN is a little bit of a "blunt" manipulation - a more time-controlled manipulation would have been more persuasive and informative. Did authors notice seizure activity in the mice? Could the authors comment on why this silencing method was chosen, and its limitations? How long did the silencing effects last for?

While using DREADDs to inhibit hippocampal INs is a "blunt" manipulation that lacks control over the effect's duration, it has the advantage of being applicable over long periods of time. The effect's duration was crucial for our experiment due to the low occurrence of SPW-R events (approximately 1-2 events per minute) in neonatal animals. To reliably detect and quantify these events, it was necessary to record for 30 to 117 minutes, with an average duration of 63 minutes. Silencing INs for such a long time using optogenetic tools is not possible without incurring in major side-effects. A potential optogenetic alternative, demonstrated by Stark et al. (2014) and Fernández-Ruiz et al. (2019), involves a closed-loop system. In this approach, the SPW-R events were detected in real time and triggered the optogenetic stimulation. The rapid online detection required for these short SPW-R events (40-60ms) is however, not possible given that the here used SPW-R detection relies on multiple steps (SPW detection, and two distinct ripple detections).

We determined the duration of silencing effects in a previous study (Kostka and Hanganu-Opatz 2023). Using the same DREADDs construct and manipulation approach, the power reduction peaked within 5 minutes after C21 injection and persisted for at least 2 hours. In the present study, we recorded the activity for 45 minutes after the administration of C21 and the silencing effect was present during the entire period.

Concerning seizure activity, in two out of nine mice we observed ultra-fast ripples approximately an hour post-injection. This effect is consistent with the effect observed in picrotoxin experiment by Stark et al. 2014, namely, that the selective suppression of optogenetically induced high frequency oscillations by picrotoxin was temporary (lasting 1 minute) and led to the subsequent emergence of higher-frequency, non-physiological epileptic oscillations (200–300 Hz 'fast ripples'). Since these ultra-fast ripples were present outside the investigated time window (i.e., 45 min after C21 administration), they did not interfere with the analysis and results of the present study.

7. Another question/comment relates to the approach taken to study the effect of ripple emergence in hippocampus on mPFC. Why was this only done using optogenetics, rather than observing the natural occurrence of ripples and studying their effects in mPFC?

We have also investigated the effects of spontaneous ripples in mPFC using the dataset of spontaneous activity simultaneously recorded from the prelimbic subdivision (PL) of the mPFC and

the hippocampal CA1 area of non-anesthetized P4-P12 mice (Fig. 8A and B). The effects of optogenetically-induced ripple-like activity are covered by Fig. 8C, D and E.

8. More generally, we assume that each animal only contributed one recording session to the dataset (one age datapoint) – could this be clarified and made more explicit in the methods section?

As suggested, we specified in Materials and Methods (lines 595-596) that each animal underwent a single recording session, thus contributing one age datapoint.

9. Copy editing changes:

- From lines 424-436 reference to figure 7 should be replaced by reference to figure 8.

We corrected (lines 453-460).

- Line 522 “strengthen” should be changed to its gerundive “strengthening”.

We corrected (line 548).

- Sentence in line 528 is mangled and should be corrected.

We corrected the sentence (lines 553-554).

Reviewer #3:

The authors of this study combined *in vivo* electrophysiology with optogenetics and chemogenetics in developing P4-P12 mice as well as *in silico* models to address the relationship between the developmental increase of inhibition and the emergence of the hippocampal ripples. By *in silico* simulation, they showed theoretically that increased inhibition to CA1 excitatory units promotes the emergence of ripples. By electrophysiological recording, they showed that the age-dependent increase of inhibition ratio in CA1 stabilized at the similar age as the emergence of spontaneous and light-induced ripples (P10), and they confirmed the stronger inhibition to pyramidal neurons in older ages. Using chemogenetics, they showed that silencing interneurons in CA1 decreased SPW-Rs rate in P11-P12 mice. Finally, the authors found that the firing rate increase in the downstream medial prefrontal cortex area was larger during spontaneous SPW-Rs and optogenetically induced CA1 high-frequency oscillations than during SPW events alone. Given all these results, the authors concluded that the development of inhibition promotes the emergence of hippocampal ripples in the third week of postnatal life. This is an interesting developmental study aiming to clarify the age at which the hippocampal CA1 ripple activity and its impact on downstream areas emerge, with implications for the emergence of memory consolidation as well as of several additional cognitive processes. I have several comments and recommendations aimed at improving the manuscript, as listed below.

We thank the reviewer for the constructive feedback and helpful comments and suggestions. We addressed them point by point below and modified the manuscript text and figures accordingly.

1. The main conclusion was largely drawn from the co-occurrence of the increased inhibition and emergence of ripples (Both on P10). This co-occurrence is more of a correlation, though the chemogenetic analysis implied some causal relationship. The authors should add some discussion about the correlational versus causal relationship between inhibition and ripple developments.

As highlighted by the reviewer, the study provides both correlative (analysis of 1/f slope and SPW-R rate) and causal (*in silico* experiments that show that augmenting inhibition in the network promotes the emergence of ripples; DREADDs experiments showing that silencing interneurons results in a reduction of the SPW-R rate) evidence of the developmental increase of inhibition and the emergence of ripples. We carefully inspected the text to avoid confusion between correlative and causal evidence and added a brief discussion of the topic (lines 123, 403-404).

2. The strength of inhibition was indirectly measured by models or population temporal firing profile, and it might be difficult to generate a direct perception of how the inhibition develops across ages. It might be more powerful if the authors can distinguish PYRs and INs from single unit level by the firing properties in individual animals and directly show the developmental profiles separately.

As highlighted by the reviewer, PYRs and INs might be differently involved during SPW-Rs, similarly to previous data from adult rodents (Stark et al. 2014). While most recorded units are putatively PYRs, due to the 60 PYRs:1 INs ratio in the HP CA1 (Bezaire and Soltesz 2013), understanding the contribution of both neuronal populations to the SPW-Rs development would be of critical relevance. The commonly used approach to classify the two types of units relies on difference in a spike waveform shape with fast-spiking (putative PV+) INs having a narrower one. However, during early

development, the IN waveform is undistinguishable from the PYR waveform (Weir et al. 2014; Chini et al., 2022). Thus, separating the contribution of PYR and IN firing at this age is not possible. Non fast-spiking INs are generally thought of not being distinguishable from PYRs based on their waveform (but see English et al., 2017 in the *adult* hippocampus for an exception).

Of note, while the temporal coordination of spiking may be affected by the level of inhibition within the network, the inhibition itself does not depend on the firing rates of PYRs and INs. The more reliable measure is the size of induced postsynaptic currents (e.g. Gan et al. 2017), a parameter not captured by firing rate analysis. Further, the approach to estimate the E-I ratio has been independently experimentally (Trakoshis et al., 2020; Chini et al., 2022) and computationally (Gao et al., 2017; Trakoshis et al., 2020; Chini et al., 2022; Nanda et al., 2023) used by several studies.

- It seems that there is a broad distribution of SPW-Rs rate and the ratio of SPWs with ripples across mice on P11 and P12 (from figure 4D, E). I would be stronger if the authors could show the individual mice differences in the ripples and inhibitions, and whether there is a correlation between ripples and inhibitions across individuals, which would support their conclusion.

To address the reviewer's comment, we explored the association between the 1/f exponent (used to measure E-I ratio) and SPW-R rates, and peak presence/absence. Animals with observed peaks showed a higher average 1/f exponent (1.923) compared to those without peaks (1.728), indicating that increased inhibition correlates with increased likelihood of detecting ripples. However, this effect did not reach statistical significance (condition effect = 0.102, 95% CI [-0.119; 0.322], $p = 0.358$, linear model) (Fig. A below). Similarly, SPW-R rate and 1/f exponent were positively correlated, yet this correlation at individual level did not reach significance (1/f exponent slope = -1.3, 95% CI [-2.692; 0.076], $p = 0.063$, linear model) (Fig. B below). Thus, ripples and inhibition weakly (i.e., below significance threshold) correlate across individuals. The lack of significance may be attributed to the high variability of the dataset and insufficient data points (Fig. A: $n=50$ P9-P12 mice, 34 mice without peaks and 16 with peaks; Fig. B: $n=23$ P11-12 mice), both drawbacks compromising the statistical model's accuracy. Furthermore, the necessity of strong enough inhibition for ripple generation does not inherently imply a direct correlation between the two variables: specifically, that higher inhibition in a network already capable of generating ripples will consistently lead to a higher ripple rate. The rate of SPW-Rs is much more likely to be controlled by other exogenous factors, such as the level of sleepiness of the animal.

- The authors discuss early development of the hippocampus (weeks 1-2) and then jump to discussing adult data and implications. There is a small but highly relevant literature on the development of ripple activity on postnatal weeks 3-4 that would be quite important to the current study, and which should be discussed. In particular, the study of Noguchi, Matsumoto, Ikegaya, *JNeurosci* 2023 that already showed the relationship between the

maturation of inhibition and of ripple activity in CA1 in vivo in mice, and the study of Farooq and Dragoi, Science 2019 that showed further maturation of ripple activity in freely moving rats, both studies of neural activity during developmental postnatal weeks 3-4.

In line with the suggestion, we added to the text a discussion of the development during the third and fourth postnatal weeks (lines 563-570) and the corresponding references.

5. Can the authors investigate other properties of ripples, such as ripple doublets and temporal timing details of the hippocampal-prefrontal interactions?

As recommended, we performed new analyses to quantify ripple doublets/multi-bout ripples as well as the dynamics of hippocampal-prefrontal interactions (fig. S7A).

Using the approach outlined by Oliva et al. 2018, we categorized ripples into three groups based on their duration: 30-50, 50-80, and 80-100 ms. A fourth group including durations > 100 ms, which has been proposed by Oliva and colleagues, was excluded from the present study, due to the ripple detection approach with a cut-off at 100 ms (fig. S4A). We did not individually identify multi-bout ripple events; the used detection method combines events closer than 30 ms, and therefore, any multi-bout events were grouped into the 50-80 or 80-100 ms categories. We observed an increase in SPW-R rates in the 30-50 ms and 50-80 ms groups with age (30-50 ms age slope = 0.060, 95% CI [0.036; 0.085], $p < 10^{-5}$, linear model; 50-80 ms age slope = 0.026, 95% CI [0.016; 0.036], $p < 10^{-6}$, linear model) (fig. S7A, left and middle), but there was no significant change in the 80-100 ms group (age slope = 0.0016, 95% CI [-0.001; 0.004], $p = 0.2348$, linear model) (fig. S7A, right). Of note, SPW-R rate in the 80-100 ms group was notably lower compared to the other groups. Since long duration ripples are known to be particularly relevant for memory, the limited rate and absence of a developmental trend in the 80-100 ms group may be attributed to the emergence and maturation of hippocampal sequences taking place during the third and fourth postnatal weeks, periods which were not captured in the experiments included in the present study.

The time course of the hippocampal-prefrontal interaction during SPWs and SPW-Rs is displayed in Figure 8B. To address the reviewer's comments and characterize the timing of HP-mPFC interaction during SPW/SPW-R events over age in more detail, we determined when the prefrontal spiking peaks in response to hippocampal SPWs. During SPW events, the peak of mPFC activity was aligned with the peak amplitude of SPWs, a property that was consistent across ages (age slope = 0.274, 95% CI [-2.556; 3.103], $p = 0.848$, linear model) (Fig. A below). During the SPW-R events, we observed a broader activation with more variable peak locations, yet still with constant over age (age slope = -9.941, 95% CI [-26.675; 6.794], $p = 0.24$, linear model) (Fig. B below).

We added the results of new analyses to the manuscript (lines 239-240) and a new supplementary figure (fig. S7A). The statistics for the new results were added to the Supplementary Table S5.

6. The computational model used in Figure 2 is used as justification for the remaining experiments. While that might have been indeed the intuition for this study, I would recommend moving the model toward the end of the manuscript (last figure) and presenting all the experimental results first.

While the order suggested by the reviewer corresponds to the structure of the first draft of the manuscript, we finally opted to position the model in Figure 2 before the primary experimental results. This decision was guided by the model's delineation of two investigated phenomena, namely, increased inhibition and the emergence of fast frequency oscillations. As mentioned in the manuscript (line 172), the model is meant to establish a mechanistic framework that drives the subsequent experimental results.

7. At some earlier ages during neurodevelopment, the features of ripples (lower frequency, fewer cycles) resemble those of other oscillation, such as the high-gamma oscillation. The authors should discuss their reasons for excluding this possible confound between ripple and high-gamma oscillations in their data.

To avoid the possible confound between ripples and high-gamma oscillations, we repeated the analysis shown in Fig. 4 (i.e., the SPW-R detection in the temporal domain) using higher frequency filters. We added four additional frequency bands: 100–200 Hz, 100-300 Hz, 80-250 Hz (used for ripple detection by Stark et al. 2014) and 100-250 Hz (recommended by Liu et al. 2022 as a ripple frequency band criterion for rodents). Comparing the SPW-R rate obtained using the original frequency filter (Fig.4D) with SPW-R rates calculated with the four additional frequency filters (fig. S5A), we detected a similar effect of age on SPW-R rate, namely, no change from P4 to P10, followed by an increase from P10 to P12 (fig. S5B). The detailed statistical results are summarized in the table below:

frequency filter	break-point location	P4-P10 slope	P10-P12 slope
80-200Hz	9.75, 95% C.I. [8.69; 10.82]	age slope = -0.0028, 95% C.I. [-0.17; 0.16], p = 0.974	age slope = 0.80, 95% C.I. [0.36; 1.23], p = 0.00096
100-200Hz	9.73, 95% C.I. [8.62; 10.84]	age slope = 0.0147, 95% C.I. [-0.06; 0.09], p = 0.691	age slope = 0.36, 95% C.I. [0.16; 0.57], p = 0.00165
100-300Hz	9.66, 95% C.I. [8.59; 10.73]	age slope = 0.0012, 95% C.I. [-0.06; 0.06], p = 0.970	age slope = 0.317, 95% C.I. [0.14; 0.49], p = 0.00098
80-250Hz	9.85, 95% C.I. [8.94; 10.76]	age slope = 0.0143, 95% C.I. [-0.05; 0.08], p = 0.664	age slope = 0.38, 95% C.I. [0.20; 0.56], p = 0.00022
100-250Hz	9.70, 95% C.I. [8.63; 10.76]	age slope = 0.0072, 95% C.I. [-0.06; 0.07], p = 0.820	age slope = 0.322, 95% C.I. [0.15; 0.49], p = 0.00096

In addition, we evaluated the effect of age and frequency band on the SPW-R rate and found that, while age alone has a significant effect on SPW-R rate, the frequency filter alone does not affect SPW-R rate in a significant way (age effect, P11 = 0.926, 95% CI [0.63; 1.23], $p < 10^{-7}$; age effect, P12 = 0.924, 95% CI [0.64; 2.21], $p < 10^{-8}$; frequency band effect, 100-200Hz = -0.023, 95% CI [-0.05; 0.008], $p = 0.147$; frequency band effect, 100-300Hz = 0.007, 95% CI [-0.02; 0.04], $p = 0.655$; frequency band effect, 80-250Hz = -0.029, 95% CI [-0.06; 0.001], $p = 0.067$; frequency band effect,

100-250Hz = -0.020, 95% CI [-0.05; 0.01], $p = 0.193$; linear mixed-effect model with interactions) (fig. S5C). Lastly, we compared the results of SPW-R detection in the temporal domain (Fig. 4D, fig. S5A) with the detection results in the frequency domain (Fig.3F) and showed that the presence of oscillatory peak in power spectrum significantly correlates with the SPW-R rate for all frequency filters (peak effect, 80-200Hz = 0.72, 95% CI [0.38; 1.06], $p < 10^{-4}$; peak effect, 100-200Hz = 0.58, 95% CI [-0.27; 0.88], $p = 0.00029$; peak effect, 100-300Hz = 0.47, 95% CI [0.21; 0.73], $p = 0.00056$; peak effect, 80-250Hz = 0.58, 95% CI [0.31; 0.86], $p = < 10^{-4}$; peak effect, 100-250Hz = 0.48, 95% CI [0.22; 0.74], $p = 0.000402$; linear model) (fig. S5D).

We added the results to the manuscript (lines 217-221, 763-764) and included a new supplementary figure (Fig. S5). The statistics for the new results were added to the Supplementary Table S5.

8. At least one study, Stark, Roux, Eichler, Buzsaki, PNAS 2015 reported induction of ripples in the adult CA1 by driving putative pyramidal neurons alone. The authors should discuss how those results compare to their conclusion that direct manipulation of inhibitory activity is necessary for artificial ripple creation.

Similar to the results reported by Stark et al., the *in vivo* optogenetics (PYR-confined light activation) and *in silico* (i.e., external input was exclusively delivered to PYR) data show that the activation of subset of pyramidal cells alone was sufficient to induce a ripple-like activity. However, this result does not imply that INs are dispensable. In an earlier study, Stark et al. reported that local blocking of GABA_A receptors with picrotoxin suppressed high frequency oscillations. Here, we complemented these data by DREADDs-induced IN silencing that leads to a decrease in the SPW-R rate.

9. Minor comments

1. The condition of mice while recording should be addressed, e.g., recording while they were resting in cages or somewhere else.

We added the missing information to the Materials and Methods section (lines 643-644).

2. Line 528 '..... throughout development (Doischer et al., 2008)' the citation format is different from other citations.

We corrected (line 554).

3. There seems to be some missing left parenthesis or redundant right parenthesis in Line 212-215, which made it difficult to understand the sentence.

We removed redundant right parentheses (lines 211-216).

4. It would be more reliable if statistical measurements can be added on figure 4G and 4H, Figure 5D, and Figure 6E.

We added the requested statistical measurements on Fig. 4G, 4H, 5D, and 6E.

We updated Figure 4 (panels G and H), figure 5D, 6E, rephrased the text (lines 236-239) and added new statistical results to the Supplementary Table S5.

REVIEWERS' COMMENTS

Reviewer #1 (Remarks to the Author):

I am happy with the changes the authors made to their paper. I think the paper can be published in its current form.

Reviewer #2 (Remarks to the Author):

Thank you for addressing all the points/comments raised. This is an excellent piece of work.

Reviewer #3 (Remarks to the Author):

The authors have fully addressed satisfactorily all my comments. As a result, the manuscript has improved on multiple angles, and I recommend it for publication in its current form.